# Forming global estimates of self-performance from local confidence

Marion Rouault [1], Peter Dayan[1,2,3,4] & Stephen M. Fleming[1,2]

Metacognition, the ability to internally evaluate our own cognitive performance, is particularly useful since many real-life decisions lack immediate feedback. While most previous studies have focused on the construction of confidence at the level of single decisions, little is known about the formation of "global" self-performance estimates (SPEs) aggregated from multiple decisions. Here, we compare the formation of SPEs in the presence and absence of feedback, testing a hypothesis that local decision confidence supports the formation of SPEs when feedback is unavailable. We reveal that humans pervasively underestimate their performance in the absence of feedback, compared to a condition with full feedback, despite objective performance being unaffected. We find that fluctuations in confidence contribute to global SPEs over and above objective accuracy and reaction times. Our findings create a bridge between a computation of local confidence and global SPEs, and support a functional role for confidence in higher-order behavioral control.

[1] Wellcome Centre for Human Neuroimaging, University College London, WC1N 3AR London, UK. [2] Max Planck UCL Centre for Computational Psychiatry and Ageing Research, University College London, WC1B 5EH London, UK. [3] Gatsby Computational Neuroscience Unit, University College London, W1T 4JG London, UK. [4] Max Planck Institute for Biological Cybernetics, 472076 Tübingen, Germany. Correspondence and requests for materials should be addressed to M.R. (email: marion.rouault@gmail.com)

Metacognition, the ability to internally evaluate our cognitive processes, is critical for adaptive behavioral control, particularly as many real-life decisions lack immediate feedback. Specifically, action outcomes can be ambiguous[1], delayed, occur only after a sequence of subsequent decisions[2], or might never occur at all[3,4]. Yet behavioral and neural evidence indicate that subjects are able to evaluate their choices online in the absence of immediate feedback, forming estimates of decision confidence[5,6] and detecting and correcting response errors[7,8]. Previous research on metacognition in both humans and animals has focused on mechanisms supporting "local" decision confidence, elicited at or around the time of a particular decision[9,10]. The formation of decision confidence is informed by stimulus evidence[11–14], reaction times[15–17], and integration of post-decision evidence[18]. Theoretically, decision confidence is proposed to correspond to a probability that a choice was correct[19], and, empirically, confidence computations are thought to depend on a network of the prefrontal and parietal brain areas[5,15,20,21].

Despite this intensive focus on the construction of "local" confidence at the level of individual decisions, it remains unclear whether and how local confidence estimates might be aggregated over time to form "global" self-performance estimates (SPEs). Global beliefs about our abilities play an important role in shaping our behavior[22], determining the goals we choose to pursue, and the motivation and effort we put into our endeavors[23,24]. Put simply, if we believe we are unable to succeed at a particular task, we may be unlikely to try in the first place. In certain situations, such beliefs may exert stronger influences on our behavior than objective performance[25,26], and distortions in global self-evaluation have been associated with various psychiatric symptoms[27,28].

However, despite their widespread behavioral influence, little is known about the mechanisms supporting the formation of global SPEs on a given task. It is likely that global SPEs incorporate external feedback when it is available. For instance, when choosing our next career move, we may learn about our self-competence over multiple evaluations of performance (such as formal appraisals), and accumulate these local evaluations into coherent global beliefs. Critically, however, when external feedback is absent, it may prove adaptive to use decision confidence as a proxy for success, aggregating local confidence estimates over longer timescales to form global SPEs.

Here, we developed a paradigm to investigate how external feedback and local decision confidence relate to global SPEs, and whether local fluctuations in decision confidence inform SPEs when external feedback is unavailable. In three experiments, human subjects played short interleaved tasks and were subsequently asked to choose the task on which they think they performed best. These task choices provided a simple behavioral assay of global SPEs. Strikingly, subjects pervasively underestimate their performance in the absence of feedback, compared with a condition with full feedback, despite objective performance being similar in the two cases. Moreover, we observe that local decision confidence influences SPEs over and above accuracy and reaction times. Our findings create a bridge between local confidence signals and global SPEs, and support a functional role for confidence in higher-order behavioral control.

## Results

**Experimental design.** In Experiment 1, conducted online, human subjects ($N = 29$) performed short learning blocks (24 trials) featuring random alternation of two tasks, which were signaled by two arbitrary color cues (Fig. 1a). Each task required a perceptual discrimination as to which of two boxes contained a higher number of dots (Fig. 1c). Two factors controlled the characteristics of each task: the task was either easy or difficult (according to the dot difference between boxes), and subjects received either veridical feedback (correct, incorrect) or no feedback following each choice (Fig. 1b). This factorial design resulted in six possible task pairings within a learning block; for instance, an Easy-

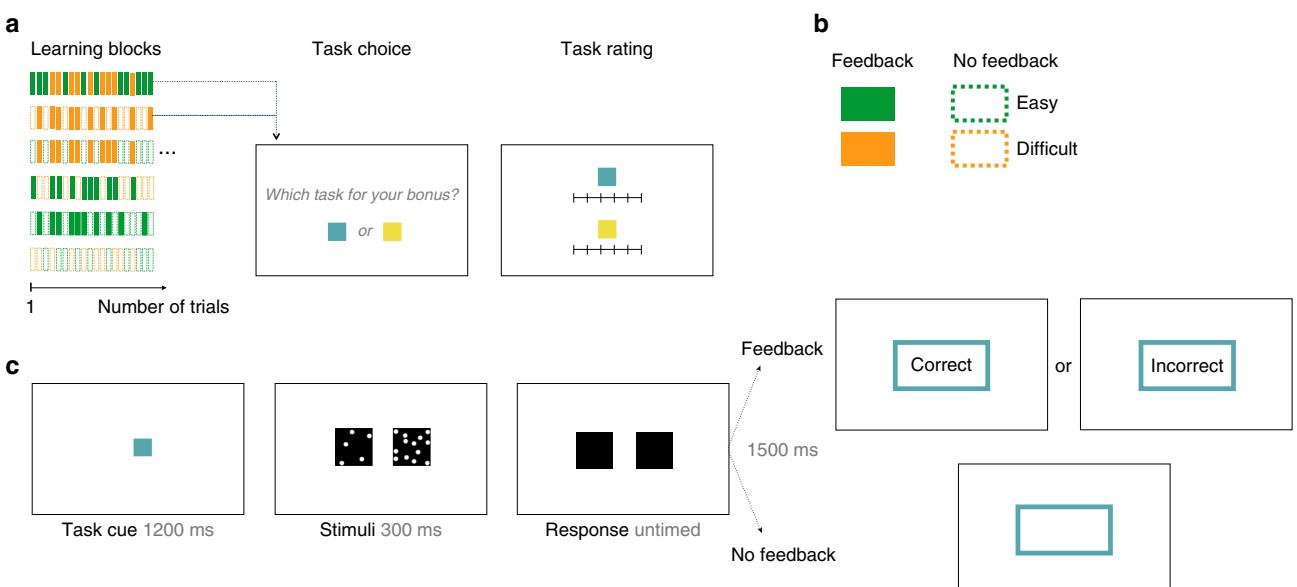

**Fig. 1** Experimental design, Experiment 1. **a** Learning blocks were composed of randomly alternating trials from two "tasks". Each task was either easy or difficult and provided feedback or no feedback (**b**), resulting in six possible task pairings (**a**). At the end of each learning block, subjects were asked to choose which task should be used to calculate a monetary bonus based on their performance at the chosen task. They were also asked to rate their overall ability at each task on a continuous rating scale. A new block ensued with two new color cues indicating two new tasks. **c** Trial structure. Each trial consisted of a perceptual judgment as to which of two boxes contained a higher number of dots. Each judgment was either easy or difficult according to the dot difference between the left and right boxes. Following their response, subjects either received veridical feedback (correct, incorrect) about their perceptual judgment or no feedback

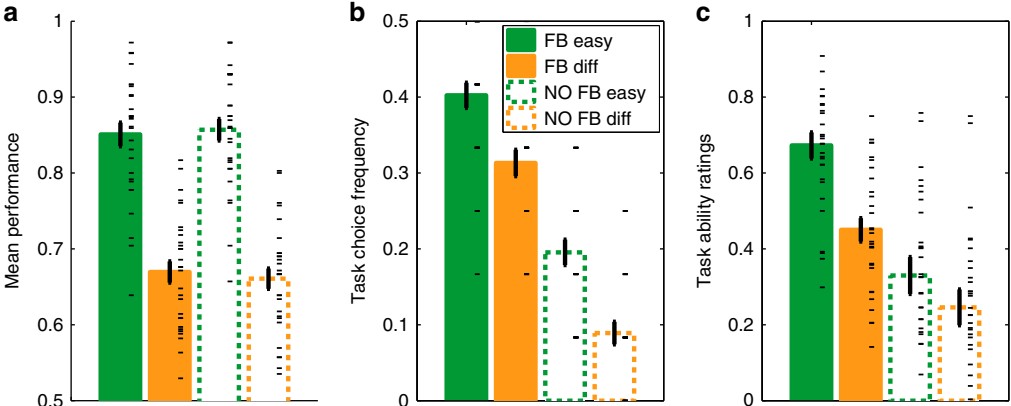

**Fig. 2** Behavioral dissociation between objective performance and SPEs in Experiment 1 ($N = 29$). **a** Performance (mean percent correct) was better for easy than difficult tasks, but was not different in tasks with and without feedback. Global self-performance estimates (SPEs) as measured directly by task choice (**b**) or indirectly via task ratings (**c**) were higher in the presence than in the absence of feedback, despite objective performance being unchanged. Error bars represent S.E.M. across subjects. Black dashes are individual data points (there are fewer task choice data points due to the limited number of blocks per subject)

Feedback task could be paired with a Difficult-Feedback task, or a Difficult-Feedback task could be paired with a Difficult-No-Feedback task, and so forth (Fig. 1a).

At the end of each learning block, subjects were asked to choose the task at which they believed they performed better (Fig. 1a). They were instructed that they would receive a monetary bonus in proportion to their performance in the chosen task, and were therefore incentivized to select the task for which they thought they performed best (see the Methods section). They were additionally asked for an overall rating of self-performance on each of the two tasks on a continuous scale (Fig. 1a and Methods). A short break ensued before the next learning block started, when two new color cues indicated two new tasks. The two end-of-block measures, task choices, and task ratings, represented our proxies for global self-performance estimates (SPEs). This experimental design allowed us to investigate how SPEs are formed when external feedback is absent—requiring subjects to rely on trial-by-trial metacognitive estimates of their performance—as compared with when feedback is available.

**Behavioral results**. We initially examined how our experimental factors affected subjects' performance on the tasks. A $2 \times 2$ ANOVA on performance revealed a main effect of Difficulty ($F (1,28) = 292.8$, $p < 10^{-15}$), but no main effect of Feedback ($F (1,28) = 0.02$, $p = 0.90$) and no interaction ($F(1,28) = 0.44$, $p = 0.51$). In particular, subjects' performance averaged 67 and 85% correct in the difficult and easy tasks, respectively (difference: $t_{28} = -17.02$, $p < 10^{-15}$); this difference in performance between difficulty levels was also present for every subject individually. Critically, within each of the two difficulty levels, objective performance was similar in the presence and absence of feedback (both difficulty levels $t_{28} < 0.58$, $p > 0.57$, BF $< 0.22$; substantial evidence for the null hypothesis), indicating that we were able to examine how feedback affects SPEs irrespective of variations in performance. A similar pattern was observed for reaction times (RTs) (main effect of Difficulty, $F(1,28) = 23.87$, $p < 10^{-4}$; no main effect of Feedback, $F(1,28) = 0.16$, $p = 0.69$ and no interaction, $F(1,28) = 0.08$, $p = 0.78$). RTs were significantly faster in easy (mean = 672 ms) as compared with difficult tasks (mean = 707 ms) ($t_{28} = 4.88$, $p < 10^{-4}$); this difference in RTs was observed in 24 out of 29 subjects. Conversely, within each of the two difficulty levels, RTs were similar in the presence and absence of

feedback (both difficulty levels $t_{28} < 0.41$, $p > 0.68$; BF $< 0.21$ i.e., substantial evidence for the null hypothesis), allowing us to examine how feedback affects SPEs independently of variations in both objective performance and RTs (Fig. 2a).

Together these analyses indicate that task difficulty affected objective performance, as expected, but performance was unaffected by the presence or absence of feedback. We next examined whether subjects were able to form global SPEs that reflected their objective performance over the course of a block. To this end, we asked whether and how the various factors of our experimental design influenced end-of-block measures of SPEs. A $2 \times 2$ ANOVA indicated a significant influence of Feedback ($F (1,28) = 112.6$, $p < 10^{-10}$) and Difficulty ($F(1,28) = 24.2$, $p < 10^{-4}$) on task choice together with an interaction between these factors ($F(1,28) = 4.5$, $p = 0.04$). Easy tasks were chosen more often than difficult tasks, even in the absence of feedback, indicating that subjects were sensitive to variations in task difficulty when making end-of-block choices (Fig. 2b). Conversely, tasks were chosen less often in the absence of feedback (Fig. 2b) despite task performance remaining similar in the presence and absence of feedback, with the interaction indicating that tasks were chosen more often in the presence of feedback, an increase which was slightly larger for difficult tasks.

Likewise, subjective ratings of overall performance were greater on easy tasks compared with difficult tasks, and again, despite task performance remaining stable in the presence and absence of feedback, performance was rated as worse in the absence of feedback (Fig. 2c). A $2 \times 2$ ANOVA revealed a significant influence of Feedback ($F(1,28) = 32.1$, $p < 10^{-5}$) and Difficulty ($F(1,28) = 51.1$, $p < 10^{-7}$) on subjective ratings together with a significant interaction ($F(1,28) = 10.5$, $p = 0.003$), indicating that tasks were rated higher in the presence of feedback and even more so for easy tasks. We note that the presence and direction of interactions between factors predicting SPEs differed across task choices and ratings (and also across experiments, see below), which may be due to the boundedness of the rating scale creating ceiling/floor effects that do not affect task choices. Importantly, however, the main effects of our Feedback and Difficulty manipulations reliably and consistently impacted SPEs across all measures.

To explore the source of these differences in SPEs, we split task choice and task ability rating data into the six types of learning blocks (Fig. 3a, b). Strikingly, subjects chose Feedback-Difficult tasks more frequently when paired with No-Feedback-Easy tasks

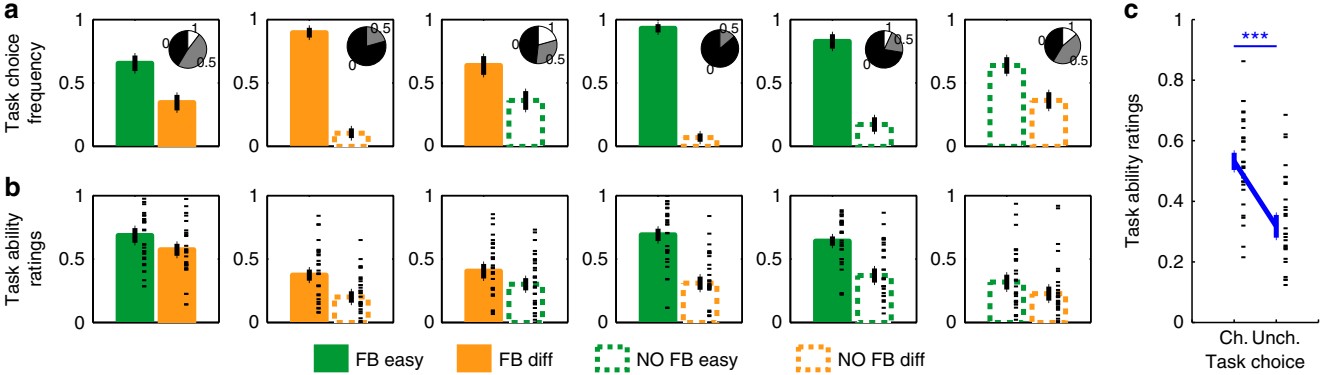

**Fig. 3** Effects of task difficulty and feedback on global SPEs in Experiment 1 ($N = 29$). **a**, **b** Task choice frequency (**a**) and task ability ratings (**b**) were visualized for the six task pairings. **a** Task choice frequencies could only take on the values 0, 0.5, or 1 due to the limited repetitions of pairing types per subject; pie charts display the fractions of subjects for whom these values were 0, 0.5 or 1 (for the right-hand bar of each plot). **b** Black dashes are individual data points. **c** Chosen tasks (Ch.) were rated more highly than unchosen tasks (Unch.), indicating consistency across our two measures of SPEs. ***$p < 0.000001$, paired $t$ test. Error bars represent S.E.M. across subjects

(28% difference in task choice frequency) (Fig. 3a, third panel), indicating a greater SPE in the former despite performing significantly better in the latter. This was also the case, albeit to a lesser extent, when examining task ratings ($t_{28} = -2.29$, $p = 0.03$) (Fig. 3b, third panel). Furthermore, subjects' task choices discriminated equally well between easy and difficult tasks in blocks where both tasks had feedback (31% difference in task choice frequency) or both had no feedback (28% difference in task choice frequency; Fig. 3a, compare first and last panels). This indicates that subjects' task choices were sensitive to variations in difficulty despite feedback being unavailable. Across all panels, subjects' task choices were more extreme than task ratings, possibly due to task choice being a binary read-out of a graded SPE (see Discussion). Critically, these differences in SPEs were not trivially explained by variations in objective performance across the six types of learning blocks: neither performance nor RTs in a given task condition differed across the different task pairings (performance, all $p > 0.36$ except for Feedback-Difficult tasks when paired with No-Feedback-Difficult vs. Feedback-Easy tasks, $p = 0.04$; RTs, all $p > 0.16$, paired $t$ tests).

Despite task choices being slightly more extreme than task ability ratings, their patterns were notably similar, with identical direction of effects in all six task pairings (Fig. 3a, b). Moreover, subjects rated chosen tasks more highly than unchosen tasks in 72% of the blocks, which reveals a high level of consistency between our two proxies for global SPEs (rating chosen vs. unchosen task: $t_{28} = 6.92$, $p < 10^{-6}$) (Fig. 3c). Accordingly, a logistic regression showed that the difference in task ratings strongly predicted task choice ($\beta = 0.24$, $p < 10^{-15}$, $r^2 = 0.41$), again indicating consistency across our two ways of operationalizing SPEs. Taken together, the results of Experiment 1 indicate that participants are sensitive to changes in task difficulty when constructing SPEs, and that self-performance is systematically underestimated in the absence of feedback as compared with when feedback is available, despite objective performance remaining stable.

**Learning dynamics**. We next sought to replicate these effects in an independent data set while additionally investigating the dynamics of SPE formation. In Experiment 2 ($N = 29$ new subjects), we varied the duration of blocks from 2 to 10 trials per task to ask how the amount of experience with each task influenced SPEs (Fig. 4a). Since task ratings followed a similar pattern as task choices in Experiment 1 (Fig. 2b, c and Fig. 3), they were omitted

in Experiment 2 (see Methods). Replicating Experiment 1, a $2 \times 2$ ANOVA indicated a main effect of Feedback ($F(1,28) = 44.5$, $p < 10^{-6}$) and Difficulty ($F(1,28) = 73.8$, $p < 10^{-8}$) on task choice in the absence of an interaction ($F(1,28) = 0.32$, $p = 0.57$). In particular, we again found that in the absence of feedback, tasks were chosen less often (Fig. 4b and Supplementary Fig. 2b), despite objective performance remaining similar with and without feedback (for both difficulty levels: both $t_{28} < 0.59$, both $p > 0.56$, both BF < 0.218; substantial evidence for the null hypothesis) (Supplementary Fig. 2a and Supplementary Notes).

To determine whether subjects were sensitive to trial-by-trial fluctuations in performance over and above variations in difficulty level, we further split blocks of different durations according to the difference in objective performance between both tasks (see Methods; note that this split could not be performed in Experiment 1 because there were only two repetitions of each task pairing per subject). As in Experiment 1, we found that the difference in objective task performance on a given block influenced SPEs over and above effects of objective difficulty (Fig. 4b). A logistic regression confirmed a significant effect of the difference in performance between tasks on end-of-block task choices (all task pairings $\beta > 6.25$, $p < 0.0005$, except when an Easy-Feedback task was paired with a Difficult-No-Feedback task (trend at $\beta = 3.97$, $p = 0.076$), presumably due to a ceiling effect (Fig. 4b, fourth panel)). Notably, in blocks where feedback was absent for both tasks, the difference in task choice frequency between easy and difficult tasks was larger when the difference in objective performance was also larger, indicating that subjects' SPEs closely followed their actual performance (Fig. 4b, last panel). Furthermore, when an Easy-No-Feedback task was paired with a Difficult-Feedback task, the difference in performance between both tasks tempered the effect of feedback (Fig. 4b, third panel): when the difference in objective task performance was smaller, subjects favored the task with external feedback, whereas when the difference was larger, SPEs followed objective difficulty levels. Last, for a given difficulty level, when the difference in performance was smaller, subjects' choices favored tasks which provided external feedback (Fig. 4b, second and fifth panels): i.e., when task performance levels are similar, external feedback is the dominant influence on SPEs.

A potential explanation of this last observation is that subjects prefer to gamble on tasks on which they are informed about their performance, due to a value-of-information effect. We consider this explanation as less likely than a true decrease in SPE in the absence of feedback (see Discussion), because the effect of

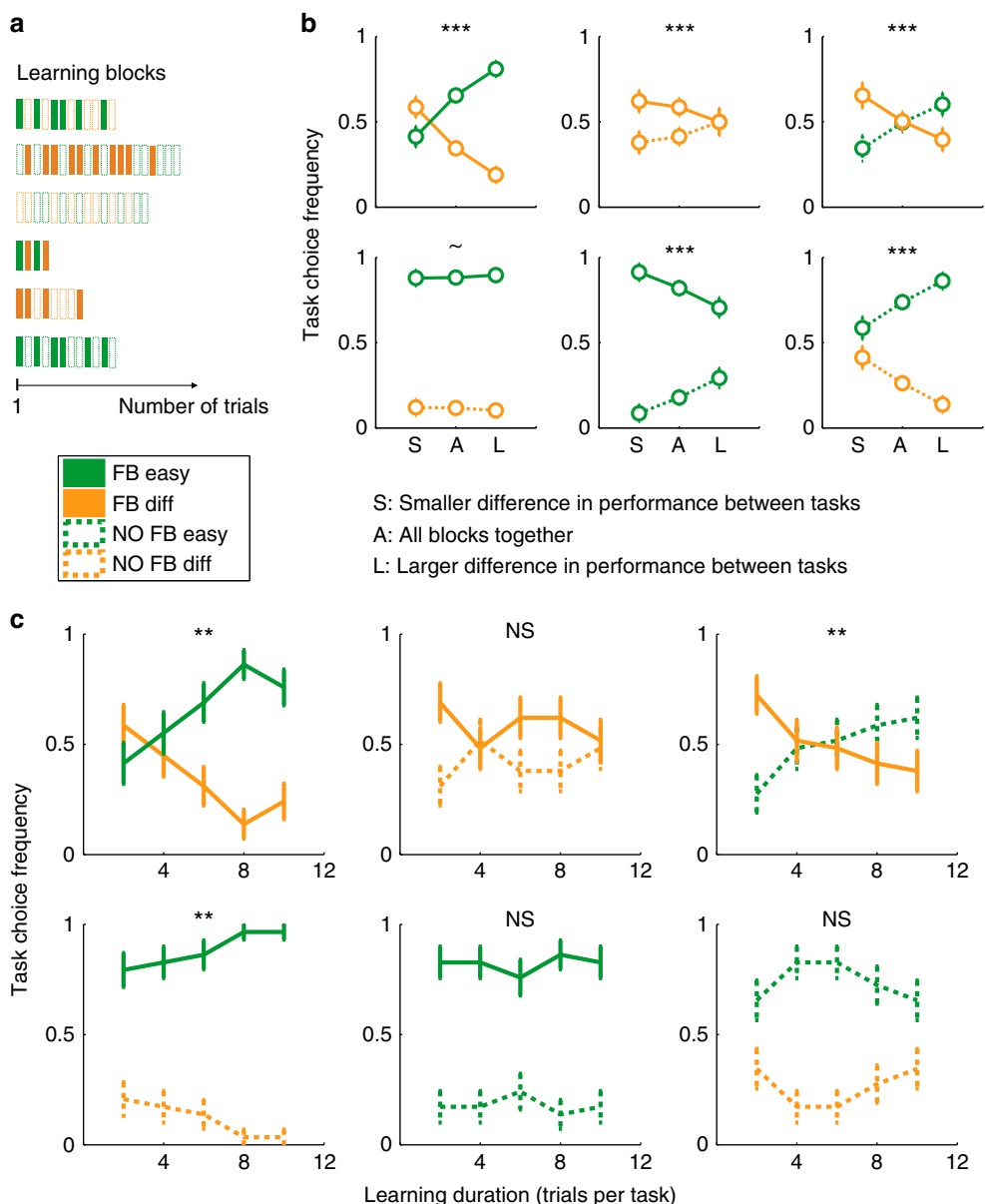

**Fig. 4** Effects of performance and learning duration on global SPEs in Experiment 2 ($N = 29$). **a** Experimental design. Block duration varied from 2 to 10 trials per task. **b** Task choice frequency in the six types of learning blocks. The central circle of each subplot represents the average task choice frequency over all blocks ["A"]. The left and right circles display the same data split into blocks with smaller ["S"] and larger ["L"] difference in objective performance between both tasks, indicating that fluctuations in local performance influenced SPEs over and above objective difficulty. ~$p = 0.076$, ***$p < 0.0005$ indicate the significance of the regression coefficient regarding the effect of the difference in task performance on task choice. **c** Task choice frequency as a function of block duration for the six task pairings. **$p < 0.01$ (resp. NS) denotes whether block duration had a significant (resp. not significant) influence on task choice (statistical significance of the logistic regression coefficient, see Methods). Error bars indicate S.E.M. across subjects. See also Supplementary Fig. 2

feedback differentially affected easy/difficult task pairings, despite these two blocks being strictly equivalent in terms of information gain (Fig. 2b, c). Importantly, these differences in SPE were not confounded by trial number: blocks with a larger difference in performance between tasks did not have a significantly greater number of trials than blocks with a smaller difference in performance (for the pairing Feedback-Difficult vs. No-Feedback-Difficult, blocks with a smaller difference in performance had a greater number of trials, $t_{28} = -2.43$, $p = 0.02$, for all other pairings, all $t_{28} < 0.65$, $p > 0.52$). Taken together, these findings indicate that subjects consider local fluctuations in their accuracy when choosing between tasks.

To establish how SPEs emerge over the course of a block, we next unpacked data according to learning block duration. Overall, block duration had a small influence on task choice frequencies (Fig. 4c). Using logistic regression (see Methods), we found that block duration significantly influenced task choice in some of the task pairings (Feedback-Easy vs. Feedback-Difficult, No-Feedback-Easy vs. Feedback-Difficult and Feedback-Easy vs. No-Feedback-Difficult; all $p < 0.01$), but not others (Feedback-Difficult vs. No-Feedback-Difficult, Feedback-Easy vs. No-Feedback-Easy and No-Feedback-Easy vs. No-Feedback-Difficult; all $p > 0.47$) (see Methods). When feedback was given on both tasks, block duration significantly influenced task choice ($p =$

0.005, statistical significance of regression coefficient) such that subjects became better at discriminating between easy and difficult tasks as more feedback was accrued (Fig. 4c, first panel). In contrast, when feedback was absent for both tasks, there was no influence of block duration ($p = 0.69$, statistical significance of regression coefficient, Fig. 4c, last panel).

**Computational modeling**. We next considered a candidate hierarchical learning model of how SPEs are constructed over the course of learning in order to explain end-of-block task choices (Fig. 3c and Supplementary Fig. 4a). The model is composed of two hierarchical levels, a perceptual module generating a perceptual choice and confidence on each trial, and a learning module updating global SPEs across trials from local decision confidence and feedback, which are then used to make task choices at the end of blocks (Supplementary Methods). An interesting property naturally emerging from this model is that over the course of trials, posterior distributions over SPEs become narrower around expected performance slightly more rapidly with feedback than without (Supplementary Fig. 4b). Model simulations provided a proof of principle that such a learning scheme is able to accommodate qualitative features of participants' learned SPEs (Supplementary Fig. 4c and Supplementary Notes). In particular, (1) the model ascribed higher SPEs to easy tasks than difficult tasks and (2) the presence of feedback led to higher SPEs than the absence of feedback, even at the expense of objective performance (Supplementary Fig. 4c, third panel). We found that the extent to which a No-Feedback-Easy task was chosen over a Feedback-Difficult task correlated across individuals with the fitted $k_{conf}$ parameter (which captures each subject's sensitivity to the input when making confidence judgments, allowing this to differ from their sensitivity $k_{ch}$ to the input when making choices) (Spearman $\rho = 0.77$, $p < 0.000005$, Pearson $\rho = 0.71$, $p < 0.0001$). This result means that participants with more sensitive local confidence estimates were also better at tracking objective difficulty in their SPEs (Supplementary Fig. 4e). However, we also found notable differences between model predictions and participants' behavior: for instance, tasks providing external feedback were chosen more frequently by participants than predicted by the model (Supplementary Fig. 4c, lower-left panels), indicating that influences beyond those considered in the current model may affect SPE construction (see Discussion).

**Establishing a direct link between local confidence and global SPEs**. Taken together, the results of Experiments 1 and 2 and associated model fits suggest that subjects trade-off external feedback against internal estimates of confidence when estimating SPEs. However, these experimental findings and corresponding model fits provided only indirect evidence that subjects were sensitive to fluctuations in internal confidence when building global SPEs. In Experiment 3, we sought to obtain direct evidence that changes in local confidence were predictive of end-of-block SPEs. To this end, a new sample of subjects ($N = 46$) were instead asked to give confidence ratings in their perceptual judgments on no-feedback trials. All other experimental features remained identical to Experiment 2 (see Methods).

Replicating Experiments 1 and 2, we found in a 2 × 2 ANOVA that both the Feedback/Confidence manipulation ($F(1,45) = 76.9$, $p < 10^{-10}$) and Difficulty ($F(1,45) = 87.6$, $p < 10^{-11}$) impacted task choice, with an interaction between these factors ($F(1,45) = 4.8$, $p = 0.03$). Tasks with external feedback were again chosen more often than tasks in which subjects rated their confidence on each trial, in the absence of feedback (Supplementary Fig. 3a). When blocks were separated according to the difference in objective performance between tasks, we again found that subjects' task

choices reflected fluctuations in local performance over and above differences in objective difficulty levels (Supplementary Fig. 3c). Overall, task choice patterns when rating confidence in Experiment 3 were similar to those found for the no-feedback condition of Experiment 2 (Supplementary Fig. 3a and 3c), suggesting these trials were treated similarly in the two Experiments. Critically and consistent with Experiments 1 and 2, performance was better in easy compared with difficult tasks (both $t_{45} > 15.9$, both $p < 10^{-19}$), but did not differ according to the feedback/confidence manipulation (both $t_{45} < 1.44$, both $p > 0.16$, both BF < 0.420; anecdotal evidence for the null hypothesis) (Supplementary Fig. 3a). However, unlike in Experiment 2, we found no significant influence of block duration on task choice (logistic regression; all $p > 0.25$, except marginally for the third pairing, $p = 0.03$) (Supplementary Fig. 3d).

We next turned to the novel aspect of Experiment 3: local ratings of confidence. Subjects gave higher confidence ratings for easy (mean = 0.82) compared with difficult (mean = 0.76) trials ($t_{45} = 8.90$, $p < 10^{-10}$), and reported greater confidence for correct (mean = 0.80) than error (mean = 0.72) trials ($t_{45} = 10.2$, $p < 10^{-12}$) (Fig. 5a and Supplementary Fig. 3b), demonstrating a degree of metacognitive sensitivity to performance fluctuations. We also computed metacognitive efficiency (meta-$d'$/$d'$), an index of the ability to discriminate between correct and incorrect trials, irrespective of performance and confidence bias[29,30] (see Methods). We found that the posterior mean for group metacognitive efficiency was 0.80, close to the SDT-optimal prediction of 1 and providing further evidence that participants' confidence ratings effectively tracked their objective performance.

We sought to directly test whether fluctuations in local confidence affected end-of-block SPEs by splitting blocks according to differences in confidence level between tasks. In line with our hypothesis, we found that the larger the difference in confidence, the more often the objectively easier task was chosen (Fig. 5c), such that SPEs were consistent with local confidence ratings. To further quantify this effect, we asked whether the difference in confidence level between tasks explained subjects' task choices over and above differences in objective performance and/or RTs. We found that fluctuations in confidence indeed explained significant variance in subjects' task choices ($\beta = 1.04$, $p < 0.0001$), over and above variations in accuracy ($\beta = -0.036$, $p = 0.81$) and RTs ($\beta = -0.009$, $p = 0.95$) (Fig. 5b). Critically, this regression model was better able to explain subjects' task choices than a reduced model, which included only the difference in accuracy and RTs as predictors (Bayesian Information Criteria: BIC = 282 for the regression model including confidence, BIC = 310 for the reduced model), confirming that local confidence fluctuations are important for explaining variance in participants' global SPEs. Moreover, in additional analyses in which regressors were orthogonalized to each other, we found virtually identical results regarding the effect of confidence on end-of-block task choices, regardless of regressor order (confDiff: all $\beta > 15.7$, all $p < 0.0001$; accDiff and rtDiff: all |$\beta$| < 2.08, all $p > 0.35$). A regression model with only confidence as a predictor (BIC = 282) was also better at predicting task choices than the reduced model with only accuracy and RTs (BIC = 310).

Finally, we considered that if subjects use local confidence to inform their SPEs, subjects who are better at discriminating between their correct and incorrect judgments should also form more accurate SPEs. In line with this hypothesis, we found that participants with higher metacognitive efficiency were also more likely to choose the easy task over the difficult task on blocks without feedback (Pearson $\rho = 0.35$, $p = 0.02$; non-parametric correlation coefficient: Spearman $\rho = 0.43$, $p = 0.003$; $N = 46$ participants) (Fig. 5d; see Supplementary Fig. 5 for correlation between global SPEs and other measures of metacognitive ability).

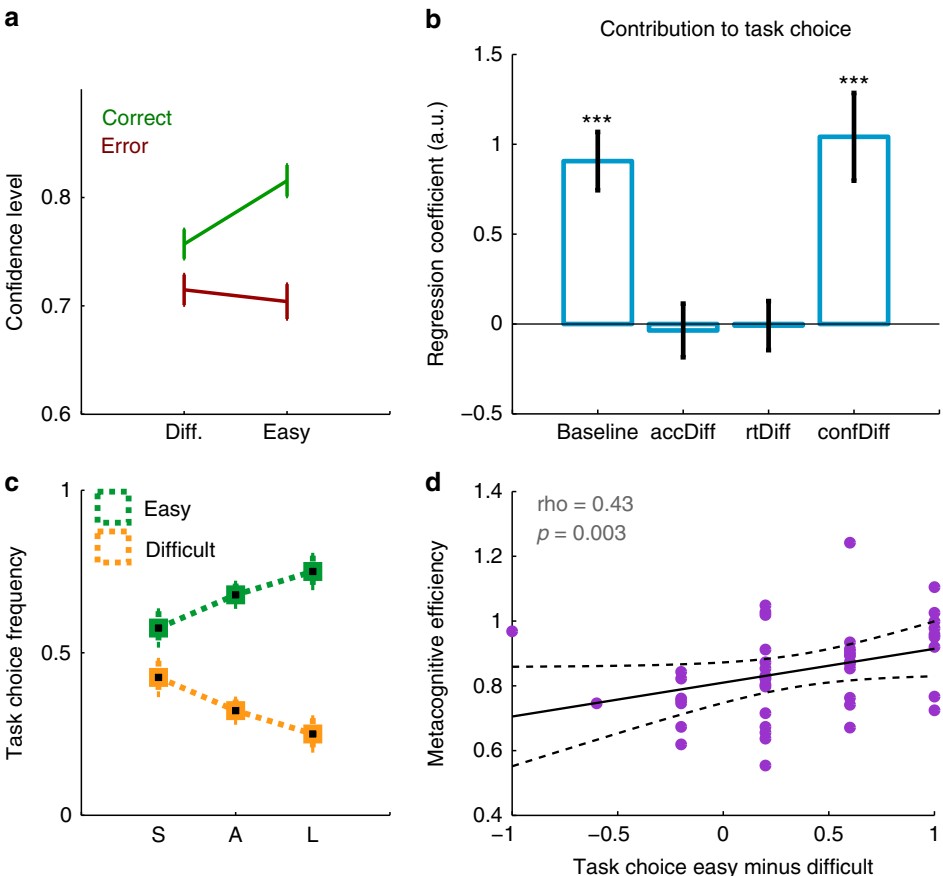

**Fig. 5** Relating local confidence to global SPEs in Experiment 3 ($N = 46$). **a** Subjects rated their confidence higher when they were correct than incorrect, and this difference was greater when the task was easier, indicating that local confidence ratings were sensitive to changes in objective performance. Error bars indicate S.E.M. across subjects. **b** Logistic regression indicating that the difference in confidence level between tasks (confDiff) explained subjects' task choices over and above a difference in accuracy (accDiff) and RTs (rtDiff). Error bars indicate S.E. over regression coefficients. ***$p < 0.0001$ indicates statistical significance of regression coefficient. **c** Fluctuations in local confidence are predictive of task choices made at the end of blocks. When the difference in mean trial-by-trial confidence between both tasks on the block was larger ("L") (respectively smaller, "S"), the difference in task choice was larger (resp. smaller). Error bars indicate S.E.M. across subjects. **d** Between-subjects scatterplot revealing that subjects with higher metacognitive efficiency were better at selecting the easiest task in end-of-block task choices. Purple dots are subjects' data, dotted lines are 95% CI. Note that task choice data are clustered in discrete levels due to a limited number of blocks per subject. See also Supplementary Fig. 3 and 5

Together, the findings of Experiment 3 support a hypothesis that fluctuations in local confidence are predictive of global SPEs and that the formation of accurate global SPEs is linked to metacognitive ability.

## Discussion

Beliefs about our abilities play a crucial role in shaping behavior. These self-performance estimates influence our choices[22], the motivation and effort we engage when pursuing our objectives[31], and are thought to be distorted in many mental health disorders[27]. However, in contrast to the recent progress made in understanding the neural and computational basis of local estimates of decision confidence[19,20,32], little is known about the formation of such global self-performance estimates (SPEs). Here, using a novel experimental design, we examined how human subjects construct SPEs over time in the presence and absence of external feedback—situations akin to many real-world contexts in which feedback is not always available.

Across three independent experiments, we observed that subjects were able to construct SPEs efficiently for short blocks of a perceptual decision task of variable difficulty. Because subjects were instructed that every new block featured two new tasks, indicated by two new color cues, they were encouraged to form their SPEs anew at the start of each block. Specifically, we found that subjects were sensitive to local fluctuations in performance and confidence within blocks when forming global SPEs. Indeed, despite stimulus evidence (i.e., difficulty level) remaining constant, variability in accuracy from block to block was reflected in subjects' task choices (Fig. 4b). This observation rules out the possibility that subjects were merely using stimulus evidence as a cue to choose between tasks at the end of blocks. Critically, we further show that fluctuations in trial-by-trial confidence were related to end-of-block SPEs, over and above effects of objective performance and reaction times (Fig. 5).

The ubiquity and automaticity of local confidence computations suggests they may be of widespread use in guiding behavior[20]. Earlier work has focused on local computations of

confidence immediately following single decisions, for instance as informed by response accuracy, stimulus evidence and reaction times in perceptual[5,32], and value-based[33] decision-making. However, the functional role(s) played by subjective confidence in decision-making have only recently received attention[17,34]. In a perceptual task offering the possibility of viewing a stimulus again before committing to a decision, a sense of confidence has been shown to mediate decisions to seek more information[35]. Likewise, perceptual confidence in a first decision modulates the speed/accuracy trade-off applied to a subsequent decision[17], and people leverage perceptual confidence when adjusting "meta-decisions" to switch environments[36] and when updating expected accuracy in a visual perceptual learning task[6]. Confidence also intervenes in cognitive offloading when deciding whether to rely on internal memories or set external reminders[37]. Here, we build on this diverse set of findings by highlighting a distinct but fundamental role for local confidence: the formation of global beliefs about self-ability. Indeed, for metacognitive evaluation to be profitable in guiding subsequent choices, confidence should aggregate over multiple instances of task experience. In keeping with this proposal, here we show that human subjects are able to form and update such global SPEs over the course of learning, and to use them for prospective decisions (such as deciding which task to do next).

Across our three experiments, we also found that the presence versus absence of feedback affected SPEs, despite objective performance remaining unaffected. Specifically, we found that subjects pervasively underestimated their performance in the absence of feedback as measured both by their task choices and ability ratings. Here, we consider three alternative explanations of this effect. First, the effect of feedback on task choices is reminiscent of a value-of-information effect:[38] subjects' choices favor tasks on which they received information about their performance. However, if this was the case, we might expect to find this effect consistently across task pairings. Instead, in Experiments 2 and 3, we found that receiving external feedback was strongly preferred in blocks where both tasks were easy (Fig. 4b, fifth panel) but only slightly preferred in blocks where both tasks were difficult (Fig. 4b, second panel), despite these two types of blocks being strictly equivalent in terms of information gain. Second, subjects may attach positive or negative valence to tasks in which they receive more positive (correct) or more negative (incorrect) outcomes, with the receipt of no feedback occupying a valence in between[39]. We note however that such a valence effect is absent in Experiment 1 (Fig. 3a, second and fifth panels), making it less likely as an overall explanation of the findings. Finally, we considered the possibility that the effect of feedback presence might be a secondary consequence of reduced uncertainty about the SPE, rather than an actual increase in SPE. Under this interpretation, subjects may have equivalent SPEs in the presence and absence of feedback, but since they would be more uncertain about their SPE in the absence of feedback, would be reluctant to gamble on their task performance when making end-of-block choices (such an effect is indeed observed in our model simulations, see Supplementary Fig. 4). However, we note that similar effects of feedback were found on both task choices and task ability ratings in Experiment 1 (Fig. 2b, c and Fig. 3a, b), and task ability ratings were also overall higher in the presence versus the absence of feedback. This observation argues against a risk-preference explanation and instead suggest that the absence of feedback leads to a genuine reduction in SPE (as assayed by subjective ratings). However, it will be of interest in future work to develop models and experimental assays that can track not only SPEs, but also the precision of SPEs (see Supplementary Fig. 4).

In all experiments, our primary probe of SPEs was a binary two-alternative forced choice. From such data, it remains uncertain whether subjects acquire SPEs in a discrete manner or gradually over the course of a learning block. For instance, there may be a particular time point within a learning sequence at which subjects commit to choosing either task in a winner-take-all fashion, and cease to update their SPEs. In support of this interpretation, we found limited evidence for an effect of block duration on task choice, suggesting that it remains possible that subjects commit to either task at an earlier time point. On the other hand, we did not observe that performance earlier in a block had a stronger effect on task choices than performance later on (Supplementary Fig. 1). SPEs may be gradually acquired, as one would expect under a Bayesian learning framework similar to the one proposed here (see Supplementary Notes). In either case, the subjective task ratings at the end of each block in Experiment 1 revealed that subjects indeed had access to a graded, parametric representation of self-performance, which followed a similar pattern to that of task choices (Fig. 3a, b). A parsimonious interpretation of these relationships is that a common latent SPE underpins both task choice and task ratings. We also found that patterns of SPEs were similar regardless of whether subjects were asked to prospectively evaluate their expected performance on a subsequent test block (Experiment 1) or retrospectively evaluate self-performance over the past learning block (Experiments 2 and 3). It is therefore possible that SPEs are aggregated from past instances in anticipation of encountering the same or a similar situation in the future[40], such as when subjects were asked to choose which task to play for a bonus in Experiment 1.

How SPEs are represented and updated at a computational and neural level remain to be determined. As an initial step in this direction, in Supplementary Notes we outline a candidate hierarchical learning model, which links local confidence to the construction of SPEs. This model includes local confidence as an internal feedback signal, formalizing the fact that the evidence available for updating SPEs is more uncertain in the absence versus presence of external feedback, as illustrated through simulations of task choices (Supplementary Fig. 4). Despite the model qualitatively capturing subjects' SPEs, more work is required to understand why subjects' task choices favor tasks with external feedback to a greater extent than predicted by the model. Due to the global nature of SPEs (as compared with trial-by-trial estimates of local confidence) and the blocked structure of our experimental design, we only had access to a limited number of data points per subject (12 task choices in Experiment 1; 30 in Experiments 2 and 3) preventing us from reliably fitting model parameter(s) and discriminating between competing models[41]. Future work could attempt to combine manipulations of local confidence with a denser sampling of SPEs to allow such model development.

The ventromedial prefrontal cortex and adjacent perigenual anterior cingulate cortex, a key hub for performance monitoring, are candidate regions for maintaining long-run SPEs[20,21,42]. Subjects may either represent SPEs in an absolute format, separately for both tasks, or in a relative format, for instance how much better they are at one task compared with another. It is possible that our experimental design with interleaved tasks may encourage a relative representation of SPEs within a block. It also remains unclear how SPEs interact with notions of expected value. In the present protocol, participants received a monetary incentive for reporting the task they thought they were better at, and thus the expected value of task choices and SPEs were correlated. Moreover, value and confidence representations are often found to rely on similar brain areas[20,33]. Conceptually, subjects might therefore represent SPEs in the frame of expected accuracy (as postulated in the present Bayesian learning scheme) or in the frame of expected value (as a reinforcement learning framework would predict; e.g[6,43].); further work is required to distinguish between these possibilities.

In Experiment 3, we found evidence that individual differences in metacognitive efficiency were related to the extent to which SPEs discriminated between easy and difficult tasks (Fig. 5d). This finding echoes a previous observation of a relationship between metacognitive efficiency and the ability to learn from predictive cues over time[44]. Metacognitive efficiency indexes the extent to which one's confidence judgments are sensitive to objective performance. To the extent to which local confidence informs SPEs, it is thus plausible that more sensitive confidence estimates translate into more accurate SPEs. However, although easy trials are more likely to be correct than difficult ones, there is only partial overlap between the determinants of metacognitive efficiency and our current measure of SPE sensitivity. More work is required to determine when and how individual variation in metacognitive efficiency influences the formation of global SPEs.

There is increasing recognition that local confidence estimates integrate multiple cues[45,46]. Interestingly, higher-order beliefs about self-ability—assayed here as SPEs—might in turn influence local judgments of confidence over and above bottom-up information obtained on individual trials[22,44,47]. This interplay of local and global confidence might be one mechanism for supporting transfer of SPEs to new tasks not encountered before[40], in a way that could prove either adaptive or maladaptive[48]. Indeed these global estimates may constitute useful internal priors on expected performance in other tasks, known as self-efficacy beliefs[22,31]. An overgeneralization of low SPEs between different tasks may even engender lowered self-esteem, leading to pervasive low mood[49], as visible in depression where subjects hold low domain-general self-efficacy beliefs[22,50]. Building models for understanding how humans learn about global self-performance from local confidence represents a first canonical step toward developing interventions for modifying this process[24,31,51].

## Methods

**Participants**. In Experiment 1, 39 human subjects were recruited online through the Amazon Mechanical Turk platform. Since we had no prior information about expected effect sizes, we based our sample size on similar studies conducted in the field of confidence and metacognition. Subjects were paid $3 plus up to $2 bonus according to their performance for a ~ 30 min experiment. They provided informed consent according to procedures approved by the UCL Research Ethics Committee (Project ID: 1260/003). The challenging nature of the perceptual stimuli, which appeared only briefly, ensured that it was impossible for subjects to perform above chance level if they were not paying careful and sustained attention during the experiment. To further ensure data quality, standard exclusion criteria were applied. Ten participants were excluded for responding at chance level and/or always selecting the same rating, leaving $N = 29$ participants (17 f/12 m, aged 22–31 and not color-blind according to self-reports) for data analysis.

In Experiment 2, 31 subjects were recruited using the same protocol as in Experiment 1. Identical exclusion criteria were applied leading to the exclusion of two subjects, leaving $N = 29$ subjects for analysis (9 f/20 m, aged 19–35).

In Experiment 3, to examine between-subjects relationships between the formation of self-performance estimates (SPEs) and metacognitive ability, 73 subjects were originally recruited online. After application of identical exclusion criteria to those used in Experiments 1 and 2, we additionally excluded subjects who failed comprehension questions about usage of the confidence scale (subjects passed if they rated "perfect" performance at least 10% greater than "chance" performance), leaving $N = 46$ subjects for analysis (16 f/30 m, aged 20–50).

Overall, our exclusion rates are consistent with recent online studies from our lab[28] and a recent meta-analysis of online studies reporting typical exclusion rates of between 3 and 37%[52]. As Experiment 3 was slightly longer, subjects' baseline pay was increased to $3.50 plus up to $2 bonus according to their performance. Subjects who participated in Experiment 1 were not permitted to take part in Experiments 2 and 3, and subjects who participated in Experiment 2 were not permitted to take part in Experiment 3.

**Experiment 1**. Subjects performed short learning blocks, which randomly interleaved two "tasks" identified by two arbitrary color cues (Fig. 1). Subjects were incentivised to learn about their own performance on each of the two tasks over the course of a learning block. Each block had 24 trials (12 trials from each task, presented in pseudo-random order). Each task required a perceptual judgment as to which of two boxes contained more dots (Fig. 1c). The difficulty level of the judgment was controlled by the difference in dot number between boxes. Any given

task (as indicated by the color cue) was either easy or difficult and provided either veridical feedback or no feedback (Fig. 1b). These four task features provided six possible pairings of tasks in the learning blocks (Fig. 1a). Each possible pairing was repeated twice, and their order of presentation was randomized within participant.

At the end of each learning block, subjects were asked to choose the task for which they thought they performed better (Fig. 1a). Specifically, they were asked to report which task they would like to perform in a short subsequent "test block" in order to gain a reward bonus. Therefore, subjects were incentivised to choose the task they thought they were better at (even if that task did not provide external feedback). This procedure aimed at revealing global self-performance estimates (SPEs), as subjects should choose the task they expect to be more successful at in the test block in order to gain maximum reward. To indicate their task choice, subjects responded with two response keys that differed from those assigned to perceptual decisions to avoid any carry-over effects. The subsequent test block contained six trials from the chosen task (not illustrated in Fig. 1). No feedback was provided during test blocks.

After the test block, subjects were asked to rate their overall performance on each of the two tasks on a rating scale ranging from 50% ("chance level") to 100% ("perfect") to obtain explicit, parametric reports of SPEs (Fig. 1a). Ratings were made with the mouse cursor and could be given anywhere on the continuous scale. Intermediate ticks for percentages 60, 70, 80, and 90% correct were indicated on the scale, but without verbal labels. Perceptual choices, task choices, and ratings were all unspeeded. After each learning block, subjects were offered a break and could resume at any time, with the next learning block featuring two new tasks cued by two new colors.

Subjects' remuneration consisted of a base payment plus a monetary bonus proportional to their performance during test blocks (see Participants). Subjects were also encouraged to give accurate task ratings (although their actual remuneration did not depend on this feature): "Your bonus winnings will depend both on your performance during the bonus [i.e., test block] trials and on the accuracy of your ratings". As data were collected online, instructions were as self-explanatory and progressive as possible, including practice trials with longer stimulus presentation times on one task (one color cue) at a time.

Each learning block featured two tasks, with each trial starting with a central color cue presented for 1200 ms, indicating which of the two tasks will be performed in the current trial (Fig. 1c). The stimuli were black boxes filled with white dots randomly positioned and presented for 300 ms, during which time subjects were unable to respond. We used two difficulty levels characterized by a constant dot difference, but the spatial configuration of the dots inside a box varied from trial to trial. One box was always half-filled (313 dots out of 625 positions), whereas the other contained 313 + 24 dots (difficult conditions) or 313 + 58 dots (easy conditions). Those levels were chosen on the basis of previous online studies in order to target performance levels of around 70 and 85% correct, respectively[28].

The location of the box that contained more dots was pseudo-randomized within a learning block with half of the trials appearing on the left, and half on the right. Subjects were asked to judge which box contained more dots and responded by pressing Z (left) or M (right) on their computer keyboard. The chosen box was highlighted for 300 ms. Afterwards, a colored rectangle (cueing the color of the current task) was presented for 1500 ms. The rectangle was either empty (on no-feedback trials) or contained the word "Correct" or "Incorrect" (on feedback trials), followed by an ITI of 600 ms. The experiment was coded in JavaScript, HTML, and CSS using jsPsych version 4.3[53], and hosted on a secure server at UCL. We ensured that subjects' browsers were in full screen mode during the whole experiment.

**Experiment 2**. To investigate how SPEs emerge over the course of learning, in Experiment 2 each block now contained either 2, 4, 6, 8, or 10 trials per task (Fig. 3a). These five possible learning durations were crossed with the six pairings of our experimental design, giving 30 blocks (= 360 trials) per participant. The dot difference for the easy conditions was changed to 313 + 60 from 313 + 58 dots, with all other experimental features remaining the same. At the end of each learning block, subjects were asked to report on which task they believed they had performed better. They were instructed that their reward bonus will depend on their average performance at the chosen task over the past learning block (instead of performance during a subsequent "test block" as in Experiment 1). Thus in Experiment 2, task choice required retrospective rather than prospective evaluation of performance, thereby generalizing the findings of Experiment 1 to metacognitive judgments with a different temporal focus. Last, given the consistency between the pattern of task ratings and task choices in Experiment 1, we decided to omit ratings in Experiment 2 to save time.

**Experiment 3**. In Experiments 1 and 2, we obtained evidence that local fluctuations in performance were linked to changes in global SPEs at the end of the block. Experiment 3 was designed to directly test whether trial-by-trial fluctuations in internal confidence were instrumental in updating global SPEs. To this end, on no feedback trials, subjects were asked to provide a confidence rating about the likelihood of their perceptual judgment being correct. The rating scale was continuous from "50% correct (chance level)" to "100% correct (perfect)", with intermediate ticks indicating 60, 70, 80, and 90% correct (without verbal labels). There was no time limit for providing confidence ratings. We did not add confidence ratings in the Feedback condition for two reasons. First, we wanted to be able to compare this

condition directly to that of Experiments 1 and 2. Second, we sought to minimize the possibility that requiring a confidence judgment might affect subsequent feedback processing in a non-trivial manner. The experimental structure and other timings remained identical to Experiment 2.

**Statistical analyses**. In Experiment 1, trials from learning blocks with reaction times (RTs) beyond three standard deviations from the mean were removed from analyses (mean = 1.59% [min = 0.69%; max = 2.78%] of trials removed across subjects). Paired $t$ tests were then performed to compare performance (mean percent correct), RTs and end-of-block task ratings between experimental conditions. To examine the influence of our experimental factors on SPEs, we carried out a $2 \times 2$ ANOVA with Feedback (present, absent) and Difficulty (easy, difficult) as factors predicting performance, task choice, and task ratings. Note that because task choice frequencies are proportions, they were transformed using a classic arcsine square-root transformation before entering the ANOVA. Task ability ratings were z-scored per subject non-parametrically (due to only 12 blocks per subject). As the absence of a difference in first-order performance (and RTs) between tasks with and without feedback is critical for interpreting differences in SPEs, we additionally conducted Bayesian paired samples $t$ tests using JASP version 0.8.1.2 with default prior values (zero-centered Cauchy distribution with a default scale of 0.707). Specifically, we evaluated the evidence in favor of the null hypothesis of no difference in performance between tasks with and without feedback, and report the corresponding Bayes factors (BF).

Since objective performance may vary even within a given difficulty level, we performed a linear regression to further quantify the influence of fluctuations in objective performance on task ratings, entering objective performance and feedback presence as taskwise regressors (two per block). Regressors were z-scored to ensure comparability of regression coefficients. In addition, we examined potential recency effects to ask whether subjects weighted all trials equally when forming global SPEs. We performed a logistic regression with accuracy (X) predicting task choice (Y) (Supplementary Fig. 1). We included four regressors (X1–X4) corresponding to the four quartiles of each block, in chronological order, with all six pairings pooled together. Subjects were treated as fixed-effects due to a limited availability of task choice data points per subject precluding the use of full random-effect models.

To provide evidence that task choice and task ability ratings were consistent proxies for SPEs, we calculated how often the chosen task was rated higher than the unchosen one, and we compared the mean ratings given for chosen and unchosen tasks (paired $t$ test). We also performed a logistic regression to examine whether the difference in ability rating between tasks predicted task choice, with subjects treated as fixed-effects due to a limited availability of task choice data points (12 blocks per subject).

To visualize the effects of learning duration in Experiment 2, end-of-block task choice frequencies were averaged across subjects for each of the six possible pairings and the five possible learning durations. Note that each data point in Fig. 4c is obtained from a different learning block, not from a series of measurements at different time points inside a block. To investigate whether learning duration exerted significant influence on task choice, separate logistic regressions were performed on each of the six task pairings. Each model was specified as Task Choice $\sim \beta_0 + \beta_1 \times$ Learning Duration and subject was treated as a fixed effect (due to only one repetition of each learning block duration per subject). In addition, to examine whether subjects' task choices took into account fluctuations in objective performance on a given learning block over and above variations in difficulty level, for each of the six pairings we split learning blocks according to the magnitude of the observed difference in performance between tasks. Specifically, we plotted task choice data for the two blocks with the smaller (resp. larger) difference in performance between tasks ("Smaller" resp. "Larger"), together with the average across all five blocks per subject (Fig. 4b). To examine quantitatively whether the difference in performance between tasks exerted a significant influence on task choice, we performed logistic regressions on each of the six task pairings. Each model was specified as Task Choice $\sim \beta_0 + \beta_1 \times$ Difference in Performance, and subject was treated as a fixed effect (again due to the availability of only one repetition of each learning block duration per task pairing per subject).

To assess use of the confidence scale in Experiment 3, we compared mean confidence (subsequently labeled "Confidence level") between correct and incorrect trials, and between easy and difficult trials (paired $t$ tests). To further establish whether subjects' confidence ratings were reliably related to objective performance, we computed metacognitive sensitivity (meta-$d$[54]). Metacognitive sensitivity is a metric derived from signal detection theory (SDT), which indicates how well subjects' confidence ratings discriminate between their correct and error trials, independent of their tendency to rate confidence high or low on the scale. When referenced to objective performance ($d'$), we can obtain a measure of metacognitive efficiency using the ratio meta-$d'/d'$. Because the meta-$d'$ framework makes the assumption of a constant signal strength across trials, we computed metacognitive efficiency separately for easy and difficult trials (corresponding to 90 trials each) and averaged the two values, which were in turn averaged at the group level. We applied a hierarchical Bayesian framework for fitting meta-$d'$, with all $\hat{R} < 1.001$ indicating satisfactory convergence[30]. We also compared the obtained metacognitive efficiency values to a classic maximum-likelihood fit[54]. Finally,

we computed a third, non-parametric measure of metacognitive ability, the area under the type 2 receiver operating curve (AUROC2), although unlike meta-$d'/d'$, this measure does not control for performance differences between conditions or subjects[54] (Supplementary Fig. 5).

To investigate whether internal fluctuations in subjective confidence were related to end-of-block SPEs, task choices in blocks where both tasks required confidence ratings were additionally split according to the difference in confidence level between both tasks (Supplementary Fig. 3b). To examine whether the difference in confidence level between tasks (confDiff) explained task choices over and above differences in accuracy (accDiff) or reaction times (rtDiff), we conducted a logistic regression on data from blocks where confidence ratings were elicited from both tasks:

$$\text{Task choice} \sim \beta_0 + \beta_1 \times \text{accDiff} + \beta_2 \times \text{rtDiff} + \beta_3 \times \text{confDiff}$$

The regressors were not orthogonalised meaning that all their common variance was placed in the residuals (Fig. 5b). Subjects were again treated as fixed-effects because we had only five data points per subject. Regressors were z-scored to ensure comparability of regression coefficients. We also ran a series of regressions as described above, but with regressors ordered and orthogonalised to each other (see Results).

Finally, we hypothesized that participants who are better at judging their own performance on a trial-by-trial basis should also form more accurate global SPEs. To this end, we asked whether individual differences in metacognitive efficiency were related to the extent to which the easy task was chosen over the difficult task. Specifically, we examined the correlation between individual metacognitive efficiency scores and task choice difference in blocks with only confidence ratings (Fig. 5d). Because there was a limited number of blocks per subject, the possible task choice proportions are clustered in discrete levels in Fig. 5d; we thus calculated both parametric (Pearson) and non-parametric (Spearman) correlation coefficients for completeness.

**Code availability**. MATLAB code for reproducing the main figures, statistical analyses and model simulations are freely available at: http://github.com/marionrouault/RouaultDayanFleming. Further requests can be addressed to the corresponding author: Marion Rouault (marion.rouault@gmail.com).

## Data availability

Behavioral summary data to reproduce the main figures and statistical analyses for all three experiments are freely available at: https://github.com/marionrouault/RouaultDayanFleming.

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

## Acknowledgements

The Wellcome Centre for Human Neuroimaging is supported by core funding from the Wellcome Trust 203147/Z/16/Z. S.M.F. is supported by a Sir Henry Dale Fellowship jointly funded by the Wellcome Trust and Royal Society (206648/Z/17/Z). P.D. was supported by the Gatsby Charitable Foundation and the Max Planck Society.

## Author contributions

M.R. and S.M.F. designed the study. M.R. programmed the experiments, collected and analyzed the data. M.R., P.D. and S.M.F. built the computational model and interpreted the results. M.R. wrote the paper with S.M.F. and P.D. providing critical revisions.

## Additional information

**Competing interests:** The authors declare no competing interests.

