## [Peer Review File · Nature Communications]

Reviewers' comments:

Reviewer #1 (Remarks to the Author):

The authors propose a new paradigm to study how global confidence estimates are built over the course of several trials. With a series of three behavioral experiments, they show that such estimates are derived from first-order performance, but also local confidence estimates. Based on these findings, they propose that local confidence estimates may be integrated over time to form global beliefs about self-ability, which may seem trivial but has never been directly quantified before. The authors nicely introduce the field, and present their work very clearly. I have no major concern regarding the empirical work, besides a few rather minor points outlined below. In my view, Experiment 1 and 2 may be considered as preparatory to Experiment 3 which is central to the study. I appreciated the efforts made to replicate results across experiments, but the contribution of experiment 1 and 2 is rather minor compared to that of Experiment 3. I would suggest to move some secondary results of Exp1/2 to SI, and instead describe the computational model in the main text.

Results

In all experiments, contrasts of interest are first presented, and only after "confirmed" by ANOVAs. I would prefer a description the other way around, starting with the ANOVA and then present pairwise comparisons where appropriate.

In all experiments, it is crucial to show that performance does not depend on feedback. In addition to the reported null effects, I recommend the use of Bayes Factors to assess how H_0 is supported.

Measures of task ability ratings were used in Experiment 1, and then dropped for Experiment 2 and 3, on the basis that they were collinear with task choice frequency. I would like to see a plot of task visibility ratings in the six task pairings (equivalent to figure 2d) in the main text. From what I could see, the two measures do not behave in the exact same way, and those subtle differences between objective and subjective measures of task performance could be discussed further.

The analysis leading to figure 3b is in my view not straightforward and difficult to follow. Instead of splitting, wouldn't it be better to use task performance as predictor in regression models, like what is applied elsewhere in the manuscript (e.g., 3c)?

In Experiment 3, local confidence estimates are measured only in the absence of feedback. In my opinion, a better alternative would have been to measure confidence in all trials, with feedback provided after the confidence report. Please justify.

In Experiment 3, an index of metacognitive efficiency is derived from local confidence estimates, but no comparison is made across experimental condition. Please discuss.

The correlation between this index and the quality of global confidence estimates (task choice easy - difficult) is fragile, as acknowledged by the authors themselves. How about other indices of metaperformance? (AROC, confidence gap, etc.)? Plus, on figure 4d, the distribution of these global confidence estimates is rather strange. Why is it clustered that way? The authors should report the correlation method they used (I suppose non-parametric considering the data distribution).

In Experiment 3, it would be interesting to assess how global confidence can be explained by sustained fluctuations during the blocks, for instance fitting autoregressive models to local confidence estimates.

A simple computational model is proposed in SI to explain the data. Although it doesn't entirely fit the results and would require further developments, I recommend to add a summary of it in the main text.

Methods: sample sizes are not justified. The fact that experiments were performed online should be mentioned in the main text.

Reviewer #2 (Remarks to the Author):

This is an interesting and well done paper about a little studied subject that is global self-performance evaluation in absence or presence of external feedback. The analysis of the data is thorough and rigorous and I believe this study gives an important contribution to the field. However some aspects of the study left me somewhat unsatisfied

Major points:

the authors evaluate global performance by asking a two-alternative forced choice in which the subjects have to choose the task in which they think they did better to receive a reward. The subjects invariably choose the task where they received a feedback. The trivial explanation of this behavior is that subjects were conservative and chose trials with feedback to maximize reward. The authors acknowledge this possibility in the discussion. In experiment one they also used another estimate of confidence that is direct confidence rating of each task. Since in experiment one confidence rating and the 2AFC gave the same result the authors decide not to repeat the direct rating in the other experiments. However this lack of direct comparison for the following experiments makes the conclusions somewhat weaker in my view

Minor points:

authors say that subjects tend to underestimate their performance in absence of feedback, but I think this is imprecise. Subjects underestimate their performance in trials with no feedback compared to trials with no feedback.

Why have the experiments been interleaved? This may have caused some confusion in the participants as opposed to two separate experiments

The colors indicating the experiments were difficult to discriminate for me, maybe some more obvious label would benefit ease of reading

Reviewer #3 (Remarks to the Author):

Rouault et al. offer an interesting and well-crafted study exploring the way in which humans establish global performance estimates based on local confidence judgments, and how this depends on difficulty and feedback. I think overall the conclusions are reasonably well-supported by the data and I see no fatal flaws, although some of my comments could motivate small tweaks to the interpretations and/or addition of one or two more caveats.

General comments:

1) I found myself frequently worrying about some critical overlooked issue only to have it addressed later on in the paper. Some of these were methods related and so I can't complain that they didn't appear sooner. But one thing I do recommend is to address what I consider the elephant in the room a bit earlier (i.e. in the Results), which is that task choice may reflect a value-of-information effect rather than SPE per se.

The authors' rebuttal to this concern, currently in Discussion, is reasonable, especially in conceding that the pattern with respect to difficulty suggests a valence effect. Indeed it seems likely that

there is an affective component driving the preference for seeking information in contexts where that information will more often be positive (more correct choices). In any event, my suggestion is to foreshadow this rebuttal in the Results so that readers do not dismiss subsequent portions of the text out of frustration that such an obvious concern is not being addressed.

2) Given the constant N_{dots} (and $diff(N_{dots})$) for a given task difficulty setting, it's a bit surprising that subjects could not ascertain the difficulty level of the task (and associated color cue) with complete certainty and then use that assessment to supersede any internal SPE they might have developed. The fact that they didn't do this might raise concerns about how well the average Mechanical Turk subject was attending to the stimuli. Assuming the author(s) piloted the task on themselves, it could be helpful to explain intuitively whether and how they believe such a strategy is difficult or impossible to employ, e.g. if it's the 300 ms presentation time that really prevents it.

3) It's hard to rely on RT in this task as a test of whether confidence informs SPE above and beyond RT, because the authors (perhaps wisely) do not assert a specific theoretical framework for linking RT and confidence, and because the range of RTs is fairly small (35 ms difference between easy and hard). However, after reading and digesting the paper as a whole I do not think this concern warrants any substantive changes, as the authors do not end up making much of the RT data. I'll leave it here though, just for completeness.

Specific/minor comments:

line 54: "global SPEs about our performance" is redundant. Suggest "global estimates of our performance," or "global SPEs"

75: discrimination judgment -> either word alone is probably fine

634: run -> ran

Why didn't the surprising finding of Fig 2d, third panel, replicate in figure 3b, top-right panel, central circle?

In the Discussion, it would be helpful to refer to specific figure panels that correspond to each data-based claim, e.g. line ~377.

Point-to-point response to reviewers' comments

We thank the reviewers for reviewing our paper. Below we describe how we revised our manuscript in response to each comment. We provide additional data and made the clarifications suggested to improve the paper.

Reviewers' comments:

Reviewer #1 (Remarks to the Author):

The authors propose a new paradigm to study how global confidence estimates are built over the course of several trials. With a series of three behavioral experiments, they show that such estimates are derived from first-order performance, but also local confidence estimates. Based on these findings, they propose that local confidence estimates may be integrated over time to form global beliefs about self-ability, which may seem trivial but has never been directly quantified before. The authors nicely introduce the field, and present their work very clearly. I have no major concern regarding the empirical work, besides a few rather minor points outlined below. In my view, Experiment 1 and 2 may be considered as preparatory to Experiment 3 which is central to the study. I appreciated the efforts made to replicate results across experiments, but the contribution of experiment 1 and 2 is rather minor compared to that of Experiment 3. I would suggest to move some secondary results of Exp1/2 to SI, and instead describe the computational model in the main text.

We thank the reviewer for their positive assessment of our paper. We have carefully considered these suggestions as described in our detailed responses below.

We agree with the reviewer that Experiment 3 is most central to the conclusions of the paper, but also think that it is important to present the evolution of the findings from Experiments 1 and 2 to Experiment 3. However, following the reviewer's suggestion, we have now streamlined the presentation of the first two Experiments, to allow the reader to move swiftly to the results of Experiment 3 (with secondary results being moved to p. 6 in Supplementary Material).

Results

In all experiments, contrasts of interest are first presented, and only after "confirmed" by ANOVAs. I would prefer a description the other way around, starting with the ANOVA and then present pairwise comparisons where appropriate.

We thank the reviewer for this suggestion and we have changed the order of presentation accordingly, presenting first the factorial ANOVAs and then the pairwise comparisons. For example, for Experiment 1, this now reads:

"We initially examined how our experimental factors affected subjects' performance on the tasks. A 2×2 ANOVA on performance revealed a main effect of Difficulty ($F(1,28)=292.8, p<10^{-15}$), but no main effect of Feedback ($F(1,28)=.02, p=.90$) and no interaction ($F(1,28)=.44, p=.51$). In particular, subjects' performance averaged 67% and 85% correct in the difficult and easy tasks respectively (difference: $t_{28}=-17.02, p<10^{-15}$); this difference in performance between difficulty levels was also present for every subject individually. Critically, within each of the two difficulty levels, objective performance (both difficulty levels $t_{28}<.58, p>.57, BF<0.22$) was similar in the presence and absence of feedback, indicating that we were able examine how feedback affects [self-performance estimates] SPEs irrespective of variations in performance." (Experiment 1, Results section, p. 6-7)

The order of presentation has similarly been changed for Experiment 2 (Results section, p. 9-10) and Experiment 3 (Results section, p. 13-14).

In all experiments, it is crucial to show that performance does not depend on feedback. In addition to the reported null effects, I recommend the use of Bayes Factors to assess how H0 is supported.

We thank the reviewer for this suggestion which we now implement for all three experiments (Methods, p. 26):

“As the absence of a difference in first-order performance (and RTs) between tasks with and without feedback is critical for interpreting differences in SPEs, we additionally conducted Bayesian paired samples *t*-tests using JASP version 0.8.1.2 with default prior values (zero-centered Cauchy distribution with a default scale of 0.707). Specifically, we evaluated the evidence in favor of the null hypothesis of no difference in performance between tasks with and without feedback, and report the associated Bayes factors (BF).”

We now report Bayes factors in the Results section; a $BF < 0.3$, indicates “positive” evidence in favor of the null hypothesis (Lee and Wagenmakers, 2013). In two experiments we found indeed substantial positive evidence ($BF < 0.3$), and in the third one anecdotal evidence ($BF < 0.4$), which overall supports a conclusion that performance did not depend on feedback:

- Experiment 1, p. 7:

“Critically, within each of the two difficulty levels, objective performance was similar in the presence and absence of feedback (both difficulty levels $t_{28} < .58$, $p > .57$, $BF < 0.22$; substantial evidence for the null hypothesis), indicating that we were able examine how feedback affects SPEs irrespective of variations in performance.”

- Experiment 2, p. 10:

“Replicating Experiment 1, we found that performance [...] was similar with and without feedback (both $t_{28} < .59$, both $p > .56$, both $BF < 0.218$; substantial evidence for the null hypothesis).”

- Experiment 3, p. 14:

“Critically and consistent with Experiments 1 and 2, performance [...] did not differ according to the feedback/confidence manipulation (both $t_{45} < 1.44$, both $p > .16$, both $BF < 0.420$; anecdotal evidence for the null hypothesis) (Supplementary Fig. 3a).”

Measures of task ability ratings were used in Experiment 1, and then dropped for Experiment 2 and 3, on the basis that they were collinear with task choice frequency. I would like to see a plot of task visibility ratings in the six task pairings (equivalent to figure 2d) in the main text. From what I could see, the two measures do not behave in the exact same way, and those subtle differences between objective and subjective measures of task performance could be discussed further.

We thank the reviewer for this suggestion. We now present the required figure displaying the task ability ratings in the six experimental pairings, together with a further comparison of task choice and task ability ratings in a new main figure (p. 9) (panel a is identical to former figure 2d):

Figure 3. a,b Task choice frequency (a) and task ability ratings (b) were visualised for the six task pairings. a, Task choice frequencies could only take on the values 0, 0.5 or 1 due to the limited repetitions of pairing types per subject; pie charts display the fractions of subjects for whom these values were 0, 0.5 or 1 (for the right-hand bar of each plot). b, Black dashes are individual data points. c, Chosen tasks (Ch.) were rated more highly than unchosen tasks (Unch.), indicating consistency across our two measures of SPEs. *** $p < .000001$. Error bars represent S.E.M across subjects.

We also expand our Results section with new analyses for assessing the similarities and differences between our two measures of SPEs, task choice and task ability ratings (p. 8-9):

“To explore the source of these differences in SPEs, we split task choice and task ability rating data into the six types of learning blocks (Fig. 3a-b). Strikingly, subjects chose Feedback-Difficult tasks more frequently when paired with No-Feedback-Easy tasks (28% difference in task choice frequency) (Fig. 3a, third panel), indicating a greater SPE in the former despite performing significantly better in the latter. This was also the case, albeit to a lesser extent, when examining task ratings ($t_{28}=-2.29$, $p=.03$) (Fig. 3b, third panel). Furthermore, subjects’ task choices discriminated equally well between easy and difficult tasks in blocks where both tasks had feedback (31% difference in task choice frequency) or both had no feedback (28% difference in task choice frequency; Fig. 3a, compare first and last panels). This indicates that subjects’ task choices were sensitive to variations in difficulty despite feedback being unavailable. Across all panels, subjects’ task choices were more extreme than task ratings, possibly due to task choice being a binary read-out of a graded SPE (see Discussion).”

“Despite task choices being slightly more extreme than task ability ratings, their patterns were notably similar, with identical direction of effects in all six task pairings (Fig. 3a-b). Moreover, subjects rated chosen tasks more highly than unchosen tasks in 72% of the blocks, which reveals a high level of consistency between our two proxies for global SPEs (rating chosen vs. unchosen task: $t_{28}=6.92$, $p<10^{-6}$) (Fig. 3c). Accordingly, a logistic regression showed that the difference in task ratings strongly predicted task choice ($\beta=0.24$, $p<10^{-15}$, $r^2=.41$), again indicating consistency across our two ways of operationalizing SPEs.”

Note that the logistic regression above replaces our previous statement about collinearity, as it is a more appropriate test for assessing a relationship between binary and continuous variables.

We also now elaborate on these findings regarding the discussion about graded vs. binary representations of SPEs. Importantly we do not argue that either measure is better, but suggest that they provide complementary windows onto SPEs (Discussion, p. 19-20):

The subjective task ratings at the end of each block in Experiment 1 revealed that subjects indeed had access to a graded, parametric representation of self-performance, which followed a similar pattern to that of task choices (Fig. 3a-b). A parsimonious interpretation of these relationships is that a common latent SPE underpins both task choice and task ratings.”

The analysis leading to figure 3b is in my view not straightforward and difficult to follow. Instead of splitting, wouldn't it be better to use task performance as predictor in regression models, like what is applied elsewhere in the manuscript (e.g., 3c)?

We thank the reviewer for prompting this analysis, which we now include (Methods, p. 28) [note that the figure concerned is now Figure 4, since we have described the new Figure 3 above]:

“To examine quantitatively whether the difference in performance between tasks exerted a significant influence on task choice, we performed logistic regressions on each of the six task pairings. Each model was specified as $\text{Task Choice} \sim \beta_0 + \beta_1 \times \text{Difference in Performance}$, and subject was treated as a fixed effect (again due to the availability of only one repetition of each learning block duration per task pairing per subject).”

We would like to keep the figure to allow visualisation of the findings, albeit with a clearer legend and colour scheme to improve readability. We also now illustrate the statistical significance of the performance predictor directly on the figure. The new regression formally confirmed the original observations (Results section, p. 11):

“As in Experiment 1, we found that the difference in objective task performance on a given block influenced SPEs over and above effects of objective difficulty (Fig. 4b). A logistic regression confirmed a significant effect of the difference in performance between tasks on end-of-block task choices (all task pairings $\beta>6.25$, $p<.0005$, except when an Easy-Feedback task was paired with a Difficult-No-Feedback task (trend at $\beta=3.97$, $p=0.076$), presumably due to a ceiling effect (Fig. 4b, fourth panel).”

Figure 4. Subjects' behavior in Experiment 2 (N=29)

a, Experimental design. Block duration varied from 2 to 10 trials per task. b, Task choice frequency in the six types of learning blocks. The central circle of each subplot represents the average task choice frequency over all blocks ['A']. The left and right circles display the same data split into blocks with smaller ['S'] and larger ['L'] difference in objective performance between both tasks, indicating that fluctuations in local performance influenced SPEs over and above objective difficulty. $\sim p=0.076$, $***p<.0005$ indicate the significance of the regression coefficient regarding the effect of the difference in task performance on task choice.

We include a similar analysis of Experiment 3, with the results illustrated in Supplementary Fig. 3c and included in the figure legend (Supplementary Material, p. 9):

*"Logistic regressions confirmed a significant influence of the difference in performance between tasks on task choices with all $\beta>3.99$, all $***p<.005$."*

In Experiment 3, local confidence estimates are measured only in the absence of feedback. In my opinion, a better alternative would have been to measure confidence in all trials, with feedback provided after the confidence report. Please justify.

We thank the reviewer for raising this interesting suggestion. We indeed considered whether to measure confidence in all trials, but decided against it for two reasons. First, we wanted to ensure that the feedback condition in Experiment 3 was similar to the same condition in Experiments 1 and 2, to allow comparison across datasets. Second, we reasoned that if we were to introduce a rating in the feedback condition, it would have to be placed before the feedback in the trial sequence. This sequence of events would have made it difficult to avoid interactions between the requirement for a confidence rating and the subsequent impact of feedback. For instance, previous studies have shown that eliciting a confidence rating influences subsequent decision-making processes (Petrucci & Baranski, 2001; 2003), and previous work from our lab and others using fMRI has revealed that requiring a metacognitive judgment leads to additional recruitment of anterior prefrontal cortex (Fleming et al., 2012; Gherman & Philiastides, 2018). In other words, reporting confidence on a scale may constitute an extra computation, which may in turn modify subsequent processing. We acknowledge that this is also potentially a problem for the no-feedback trials, but given that on these trials, there is no subsequent event following the confidence rating, we think it is less of a worry.

We now summarise this justification of our experimental choice in the Methods section (p. 26):

"We did not add confidence ratings in the Feedback condition for two reasons. First, we wanted to be able to compare this condition directly to that of Experiments 1 and 2. Second, we sought to minimize the possibility that requiring a confidence judgment might affect subsequent feedback processing in a non-trivial manner."

In Experiment 3, an index of metacognitive efficiency is derived from local confidence estimates, but no comparison is made across experimental conditions. Please discuss.

We thank the reviewer for this suggestion, and have now performed a comparison of various metacognitive ability indices in easy and difficult conditions.

We found that metacognitive ability was higher when estimated on easy as compared to difficult trials (AUROC2: $t_{45}=8.94$, $p<10^{-10}$, HMeta- d'/d' : 95% HDI of the difference [0.03-0.45]). This is consistent with subjects being less able to discriminate between their correct and incorrect trials in difficult tasks (Fig. 5a) and/or a nonlinearity in the relationship between d' and meta- d' . Ongoing work in our lab is exploring this issue and initial findings suggest that this difference between easy and difficult conditions may depend on whether trials are blocked or interleaved. We would therefore prefer not to expand further on these results until this picture becomes clearer, since this finding is tangential to the main claims of the current study.

The correlation between this index and the quality of global confidence estimates (task choice easy - difficult) is fragile, as acknowledged by the authors themselves. How about other indices of metaperformance? (AROC, confidence gap, etc.)? Plus, on figure 4d, the distribution of these global confidence estimates is rather strange. Why is it clustered that way? The authors should report the correlation method they used (I suppose non-parametric considering the data distribution).

We thank the reviewer for prompting us to explore the relationship between individual metacognitive ability and global self-performance estimates further. We now focus on three indices of metaperformance that are the best bias-free metrics presently available: meta- d' (estimated both using single-subject maximum likelihood (MLE) and within a hierarchical model) and AUROC2 (Fleming & Lau, 2014).

In the course of this new analysis, we found that parameter recovery was unstable when estimating metacognitive efficiency (meta- d'/d') with the standard MLE approach. As explained in Supplementary Methods (p. 7):

“To further assess the reliability of the MLE estimates of meta- d' , we performed additional parameter recovery simulations. Specifically, we generated confidence rating data from $N=46$ simulated subjects with 90 trials per subject following the procedures outlined in ³⁰. The group metacognitive efficiency was set to 0.8, and individual subject meta- d'/d' values were sampled from a Gaussian distribution centered on $d'=1.55$ (the mean d' value we observed for Experiment 3 data across easy and difficult conditions) with $SD=0.5$. We sampled confidence rating counts for known meta- d'/d' values using the `metad_sim` function from the HMeta- d toolbox (<https://github.com/metacoglab/HMeta-d>), keeping confidence rating criteria fixed across subjects. We observed that the ground truth meta- d' values were recovered much more accurately when using the hierarchical compared to the MLE fits (Supplementary Fig. 5d).”

Notably, the estimation procedure for meta- d' that shows the most stable parameter recovery – HMeta- d – is also the measure that also shows a significant relationship with task choice, a finding that is now supported by a similar relationship observed with the model-free AUROC2 measure. We therefore now omit the caveat about the lack of significant relationship when using the MLE approach in Discussion due to concerns over the reliability of this measure. For completeness, we instead report in Supplementary Fig. 5a-c (copied below) the correlation between global SPEs and the three indices of metacognitive ability.

In relation to the reviewer's second concern, the reason that task choice data is clustered is due to a limited number of blocks per subject, such that differences in task choice proportions can only take on one of a discrete set of possibilities. Thus it is not the case that the data is ordinal per se – it is continuous in nature, but is discretised due to the experimental design. However, we note that parametric and non-parametric correlations provided virtually identical results for the three indices of metacognitive ability, confirming the robustness of the relationship between metacognition (as estimated by HMeta- d and AUROC2) and global SPEs:

- H-Meta- d' :
Pearson $\rho=.35$, $p=.02$ (parametric correlation coefficient)
Spearman $\rho=.43$, $p=.003$ (non-parametric correlation coefficient)

- AUROC2:
Pearson $\rho=.44$, $p=.0024$
Spearman $\rho=.45$, $p=.0016$

- MLE Meta- d' :
Pearson $\rho=.12$, $p=.45$
Spearman $\rho=.21$, $p=.18$

We have updated our methods and results sections accordingly (p. 16):

“We also considered that if subjects use local confidence to inform their SPEs, subjects who are better at discriminating between their correct and incorrect judgments would also form more accurate SPEs. In line with this hypothesis, we found that participants with higher metacognitive efficiency were also more likely to choose the easy task over the difficult task on blocks without feedback (Pearson $\rho=.35$, $p=.02$; non-parametric correlation coefficient: Spearman $\rho=.43$, $p=.003$; $N=46$ participants) (Fig. 5d; see Supplementary Fig. 5 for correlation between global SPEs and other measures of metacognitive ability).”

“Note that task choice data is clustered in discrete levels due to a limited number of blocks per subject.” (legend Fig. 5, p. 15)

“Because there was a limited number of blocks per subject, the possible task choice proportions are clustered in discrete levels in Fig. 5d; we thus calculated both parametric (Pearson) and non-parametric (Spearman) correlation coefficients for completeness.” (Methods, p. 29)

Supplementary Figure 5. Relationship between three measures of metacognitive ability and global SPEs a-c. Between-subjects correlations between metacognitive ability and task choices. Purple dots are subjects' data ($N=46$), dotted lines are 95% CI. Both metacognitive efficiency ($meta-d'/d'$) when estimated hierarchically (c, identical to Fig. 5d) (Pearson $\rho=.35$, $p=.02$, Spearman $\rho=.43$, $p=.003$) and metacognitive ability estimated as the area under the type 2 receiver operating curve (AUROC2) (b) (Pearson $\rho=.44$, $p=.0024$, Spearman $\rho=.45$, $p=.0016$) showed that subjects with better metacognition were also better at selecting the easiest of both tasks in end-of-block task choices, whereas there was no significant association for metacognitive efficiency as estimated using a maximum likelihood fit (MLE) (a) (Pearson $\rho=.12$, $p=.45$, Spearman $\rho=.21$, $p=.18$). d. Parameter recovery indicating that $meta-d'$ estimation was more reliable when estimated hierarchically as compared to MLE (see Supplementary Methods). This difference in reliability indicates more credence should be given to the correlation identified via the hierarchical fit in panel c.

In Experiment 3, it would be interesting to assess how global confidence can be explained by sustained fluctuations during the blocks, for instance fitting autoregressive models to local confidence estimates.

We agree with the reviewer that there may be fluctuations in local confidence ratings over a slower timescale during the blocks. Previous studies have highlighted similar sequential effects in perceptual-decision making (Frund et al., 2014; Braun et al., 2018) and in confidence ratings (Rahnev et al., 2015). However, our design is not well-suited for addressing this question as the two tasks were interleaved across trials, such that a majority of blocks contained feedback trials alternating with confidence trials. We would therefore prefer to leave the interesting question of how sequential dependencies in local confidence affect global SPEs for future study.

A simple computational model is proposed in SI to explain the data. Although it doesn't entirely fit the results and would require further developments, I recommend to add a summary of it in the main text.

We thank the reviewer for this suggestion, and now add a summary of the key features of the model to our Results section (p. 12-13):

"We next considered a candidate hierarchical learning model of how SPEs are constructed over the course of learning in order to explain end-of-block task choices (Fig. 3c and Supplementary Fig 4a). The model is composed of two hierarchical levels, a perceptual module generating a perceptual choice and confidence on each trial, and a learning module updating global SPEs across trials from local decision confidence and feedback, which are then used to make task choices at the end of blocks (Supplementary Methods). An interesting property naturally emerging from this model is that over the course of trials, posterior distributions over SPEs become narrower around expected performance slightly more rapidly with feedback than without (Supplementary Fig. 4b). Model simulations provided a proof of principle that such a learning scheme is able to accommodate qualitative features of participants' learned SPEs (Supplementary Fig. 4c and Supplementary Results). In particular, (1) the model ascribed higher SPEs to easy tasks than difficult tasks and (2) the presence of feedback led to higher SPEs than the absence of feedback, even at the expense of objective performance (Supplementary Fig. 4c, third panel). We found that the extent to which a No-Feedback-Easy task was chosen over a Feedback-Difficult task correlated across individuals with the fitted k_{conf} parameter (which captures each subject's sensitivity to the input when making confidence judgements, allowing this to differ from their sensitivity k_{ch} to the input when making choices) (Spearman $\rho=.77$, $p<.000005$, Pearson $\rho=.71$, $p<.0001$). This result means that participants with more sensitive local confidence estimates were also better at tracking objective difficulty in their SPEs (Supplementary Fig. 4e). However we also found notable differences between model predictions and participants' behavior: for instance, tasks providing external feedback were chosen more frequently by participants than predicted by the model (Supplementary Fig. 4c, lower-left panels), indicating that influences beyond those considered in the current model may affect SPE construction (see Discussion). Taken together, the results of Experiments 1 and 2 and associated model fits suggest that subjects trade-off external feedback against internal estimates of confidence when estimating SPEs."

Methods: sample sizes are not justified. The fact that experiments were performed online should be mentioned in the main text.

As the current experiment required development of a new paradigm, we had no prior information about expected effect sizes to base our sample size upon. Instead, we set out to conduct a series of experiments, each of similar sample size to previous work in the field of confidence and metacognition, to ensure the reproducibility of key results. Experiment 3 had more participants as we were also interested in examining individual differences in metacognitive ability. We now make this clear in the main text:

"Since we had no prior information about expected effect sizes, we based our sample size on similar studies conducted in the field of confidence and metacognition." (Methods section, p. 22)

"To examine between-subjects relationships between the formation of self-performance estimates (SPEs) and metacognitive ability, 73 subjects were originally recruited". (Methods section, p. 22)

Moreover, we made sure to replicate our findings across three separate experiments on separate groups of subjects, thereby ensuring the robustness of our conclusions.

We now mention in the first sentence of the Results section that the experiments were conducted online (p. 5).

Thank you for your review.

Reviewer #2 (Remarks to the Author):

This is an interesting and well done paper about a little studied subject that is global self-performance evaluation in absence or presence of external feedback. The analysis of the data is thorough and rigorous and I believe this study gives an important contribution to the field. However some aspects of the study left me somewhat unsatisfied.

We thank the reviewer for their positive assessment of our paper. We have carefully considered their suggestions as described in our detailed responses below.

Major points:

the authors evaluate global performance by asking a two-alternative forced choice in which the subjects have to choose the task in which they think they did better to receive a reward. The subjects invariably choose the task where they received a feedback. The trivial explanation of this behavior is that subjects were conservative and chose trials with feedback to maximize reward. The authors acknowledge this possibility in the discussion. In experiment one they also used another estimate of confidence that is direct confidence rating of each task. Since in experiment one confidence rating and the 2AFC gave the same result the authors decide not to repeat the direct rating in the other experiments. However this lack of direct comparison for the following experiments makes the conclusions somewhat weaker in my view.

We thank the reviewer for raising this important point. We note however that subjects did *not* invariably choose the task where they received a feedback (Fig. 2b). Instead, our central finding is that subjects traded-off the two experimental factors, Difficulty and Feedback, when choosing between tasks at the end of blocks (Fig. 2b and Fig. 3a). Moreover, we note that a strategy of always choosing feedback does not maximize reward, because subjects were instructed that they will be rewarded on the basis of their actual performance.

We now emphasize this point in the Methods section (p. 23):

“Therefore subjects were incentivised to choose the task they thought they were better at (even if that task did not provide external feedback). This procedure aimed at revealing global self-performance estimates (SPEs), as subjects should choose the task they expect to be more successful at [...] in order to gain maximum reward.”

Moreover, we performed additional analyses to provide further support for the reliability of our 2AFC task choice in assaying global self-performance estimates (SPEs), and to provide evidence that the two measures, task ratings and task choice, are largely consistent.

First, we now present task ability ratings in the six experimental pairings for further comparison in a new main figure, in which we also show that critically subjects gave higher task ability ratings for chosen than unchosen tasks (p. 9):

Figure 3. *a,b* Task choice frequency (*a*) and task ability ratings (*b*) were visualised for the six task pairings. *a*, Task choice frequencies could only take on the values 0, 0.5 or 1 due to the limited repetitions of pairing types per subject; pie charts display the fractions of subjects for whom these values were 0, 0.5 or 1 (for the right-hand bar of each plot). *b*, Black dashes are individual data points. *c*, Chosen tasks (Ch.) were rated more highly than unchosen tasks (Unch.), indicating consistency across our two measures of SPEs. *** $p < .000001$. Error bars represent S.E.M across subjects.

Second, we expand the part of our Results section that reports the similarities between these two measures of SPEs (p. 8-9):

“Despite task choices being slightly more extreme than task ability ratings, their patterns were notably similar, with identical direction of effects in all six task pairings (Fig. 3a-b). Moreover, subjects rated chosen tasks higher (resp. higher or equal) than unchosen tasks in 72% (resp. 89%) of the blocks, which reveals a high level of consistency between our two proxies for global SPEs (rating chosen vs. unchosen task: $t_{28} = 6.92$, $p < 10^{-6}$) (Fig. 3c). Accordingly, a logistic regression showed that the difference in task ratings strongly predicted task choice ($\beta = 0.24$, $p < 10^{-15}$, $r^2 = .41$), again indicating consistency across our two ways of operationalizing SPEs.”

We also elaborate on these findings in our discussion about representations of SPEs. Note that we do not argue that either measure for assessing SPEs is better, but that both measures provide complementary windows onto SPE representations (Discussion, p. 19-20):

The subjective task ratings at the end of each block in Experiment 1 revealed that subjects indeed had access to a graded, parametric representation of self-performance, which followed a similar pattern to that of task choices (Fig. 3a-b). A parsimonious interpretation of these relationships is that a common latent SPE underpins both task choice and task ratings.”

Minor points:

authors say that subjects tend to underestimate their performance in absence of feedback, but I think this is imprecise. Subjects underestimate their performance in trials with no feedback compared to trials with no feedback.

We agree with the reviewer that the observation of performance underestimation in the absence of feedback is relative, and now make sure to highlight this point in our Results section (p. 9):

“Self-performance is systematically underestimated in the absence of feedback as compared to when feedback is available, despite objective performance remaining stable.”

This point is also highlighted in our Discussion section (p. 20):

“It is possible that our experimental design with interleaved tasks may encourage a relative representation of SPEs within a block.”

Why have the experiments been interleaved? This may have caused some confusion in the participants as opposed to two separate experiments

We thank the reviewer for raising this possibility, but wish to stress however the *experiments* were not interleaved – Experiments 1-3 were conducted on separate groups of subjects, as described in Methods. We suspect that the reviewer is referring to the fact that *tasks* were indeed interleaved within a given experiment. We think that our findings make it unlikely that participants were confused, since they provided global self-performance estimates that were sensitive to fluctuations in task performance and to our two experimental factors, difficulty level and feedback. We also note that difficulty level and feedback factors were constant within a given task during each block, and each of the two tasks was clearly cued at the beginning of each trial. If we had not interleaved the tasks, then we would likely have suffered from debilitating order effects. However, as we acknowledge in response to the previous point, such interleaving may have encouraged SPE representations to be relative to each other within a block.

The colors indicating the experiments were difficult to discriminate for me, maybe some more obvious label would benefit ease of reading

Thank you - we have now updated all figures for improved readability.

Thank you for your review.

Reviewer #3 (Remarks to the Author):

Rouault et al. offer an interesting and well-crafted study exploring the way in which humans establish global performance estimates based on local confidence judgments, and how this depends on difficulty and feedback. I think overall the conclusions are reasonably well-supported by the data and I see no fatal flaws, although some of my comments could motivate small tweaks to the interpretations and/or addition of one or two more caveats.

We thank the reviewer for their positive assessment of our paper.

General comments:

1) I found myself frequently worrying about some critical overlooked issue only to have it addressed later on in the paper. Some of these were methods related and so I can't complain that they didn't appear sooner. But one thing I do recommend is to address what I consider the elephant in the room a bit earlier (i.e. in the Results), which is that task choice may reflect a value-of-information effect rather than SPE per se.

The authors' rebuttal to this concern, currently in Discussion, is reasonable, especially in conceding that the pattern with respect to difficulty suggests a valence effect. Indeed it seems likely that there is an affective component driving the preference for seeking information in contexts where that information will more often be positive (more correct choices). In any event, my suggestion is to foreshadow this rebuttal in the Results so that readers do not dismiss subsequent portions of the text out of frustration that such an obvious concern is not being addressed.

We agree with the reviewer that the possibility that the task choice pattern may reflect a value-of-information effect is an important point to consider early on in the manuscript, and we now acknowledge this possibility in our Results section (p. 11):

"A potential explanation of this last observation is that subjects prefer to gamble on tasks on which they are informed about their performance, due to a value-of-information effect. We consider this explanation as less likely than a true decrease in SPE in the absence of feedback (see Discussion), because the effect of feedback differentially affected easy/difficult task pairings despite these two types of block being strictly equivalent in terms of information gain (Fig. 2b-c)."

We also note that the valence aspect of this value-of-information effect, i.e. the possibility that an affective component drives a preference for seeking information in contexts where that information will more often be positive (more correct choices), seems to be less robust; it is only present in Experiments 2 and 3 but not in Experiment 1, as visible in Fig. 3a (former Fig. 2d) (lack of difference between second and fifth panels):

Figure 3. *a,b* Task choice frequency (*a*) and task ability ratings (*b*) were visualised for the six task pairings. *a*, Task choice frequencies could only take on the values 0, 0.5 or 1 due to the limited repetitions of pairing types per subject; pie charts display the fractions of subjects for whom these values were 0, 0.5 or 1 (for the right-hand bar of each plot). *b*, Black dashes are individual data points. *c*, Chosen tasks (Ch.) were rated more highly than unchosen tasks (Unch.), indicating consistency across our two measures of SPEs. *** $p < .000001$. Error bars represent S.E.M across subjects.

We have additionally expanded our Discussion point about the value-of-information effect to incorporate these observations (p. 18-19):

“Across our three experiments, we also found that the presence vs. absence of feedback affected SPEs despite objective performance remaining unaffected. Specifically, we found that subjects pervasively underestimated their performance in the absence of feedback as measured both by their task choices and ability ratings. Here we consider three alternative explanations of this effect. First, the effect of feedback on task choices is reminiscent of a value-of-information effect³⁸: subjects’ choices favour tasks on which they received information about their performance. However, if this was the case, we might expect to find this effect consistently across task pairings. Instead, in Experiments 2 and 3, we found that receiving external feedback was strongly preferred in blocks where both tasks were easy (Fig. 4b, fifth panel) but only slightly preferred in blocks where both tasks were difficult (Fig. 4b, second panel), despite these two types of block being strictly equivalent in terms of information gain. Second, subjects may attach positive or negative valence to tasks in which they receive more positive (correct) or negative (incorrect) outcomes, with the receipt of no feedback occupying a valence in between³⁹. We note however that such a valence effect was absent in Experiment 1 (Fig. 3a, second and fifth panels), making it less likely as an overall explanation of the findings. Finally, we considered the possibility that the effect of feedback presence might be a secondary consequence of reduced uncertainty about the SPE, rather than an actual increase in SPE. Under this interpretation, subjects may have equivalent SPEs in the presence and absence of feedback, but since they would be more uncertain about their SPE in the absence of feedback, would be reluctant to gamble on their task performance when making end-of-block choices (such an effect is indeed observed in our model simulations, see Supplementary Fig. 4). However, we note that similar effects of feedback were found on both task choices and task ability ratings in Experiment 1 (Fig. 2b-c and Fig. 3a-b), and task ability ratings were also overall higher in the presence versus the absence of feedback. This observation argues against a risk-preference explanation and instead suggest that the absence of feedback leads to a genuine reduction in SPE (as assayed by subjective ratings).”

2) Given the constant Ndots (and diff(nDots)) for a given task difficulty setting, it’s a bit surprising that subjects could not ascertain the difficulty level of the task (and associated color cue) with complete certainty and then use that assessment to supersede any internal SPE they might have developed. The fact that they didn’t do this might raise concerns about how well the average Mechanical Turk subject was attending to the stimuli. Assuming the author(s) piloted the task on themselves, it could be helpful to explain intuitively whether and how they believe such a strategy is difficult or impossible to employ, e.g. if it’s the 300 ms presentation time that really prevents it.

We thank the reviewer for raising this important point. During lengthy piloting on ourselves, we in fact found it surprisingly hard to identify whether a stimulus was an easy or a difficult one. This is perhaps to be expected given that the generative distributions for easy and difficult stimuli (as inferred from the d' values in the two conditions) overlap considerably, and in light of previous findings that subjective confidence estimates

marginalise over possible stimulus strengths, at least under near-threshold conditions¹⁵. We now add a section of Supplementary Material to expand on this issue:

“We note that for the range of d' values we observed in our participants, the distributions of internal evidence generated from easy and difficult stimuli are expected to overlap considerably. This precludes straightforward inference about the difficulty of individual stimuli.” (Supplementary Material, p. 3)

We also think that several of our findings make it unlikely that subjects could do the task online without fully attending to the stimuli, for instance the fact that they responded significantly better than chance level, and that they provided global self-performance estimates (SPEs) that were sensitive to fluctuations in stimulus difficulty:

“The challenging nature of the perceptual stimuli, which appeared only briefly, ensured that it was impossible for subjects to perform above chance level if they were not paying careful and sustained attention during the experiment.” (Methods section, p. 22)

3) It's hard to rely on RT in this task as a test of whether confidence informs SPE above and beyond RT, because the authors (perhaps wisely) do not assert a specific theoretical framework for linking RT and confidence, and because the range of RTs is fairly small (35 ms difference between easy and hard). However, after reading and digesting the paper as a whole I do not think this concern warrants any substantive changes, as the authors do not end up making much of the RT data. I'll leave it here though, just for completeness.

We agree with the reviewer that the link between RT and confidence is not yet fully understood. One influential framework proposes that RTs inform a computation of confidence (Kiani & Shadlen, 2014), whereas other authors have suggested that confidence influences RTs, allowing subjects to slow down in order to gather more evidence when confidence is low (Dotan, Meyniel & Dehaene, 2018). Here we decided not to exploit RT data further due to a small difference between easy and difficult conditions, as pointed out by the reviewer, which we suspect may be due to RT data collected online being noisier than similar data collected in the lab.

Specific/minor comments:

line 54: “global SPEs about our performance” is redundant. Suggest “global estimates of our performance,” or “global SPEs”

Thank you, we have updated this.

**75: discrimination judgment -> either word alone is probably fine
634: run -> ran**

Thank you, we have corrected these.

Why didn't the surprising finding of Fig 2d, third panel, replicate in figure 3b, top-right panel, central circle?

This discrepancy is presumably due to the level of difficulty for easy tasks being slightly different in Experiment 2 and 3 as compared to Experiment 1 (Methods section, p. 24):

“The dot difference for the easy conditions was changed to 313+60 from 313+58 dots, with all other experimental features remaining the same.”

The original motivation for making the easy tasks a little easier was to examine whether this finding (third panel) would replicate when the difference in difficulty levels was made more extreme, making it less likely that subjects would estimate themselves to be better in a task which was objectively more difficult and in which they performed less well. This is indeed what we observed in Experiment 2, again indicating that subjects traded-off the two experimental factors, difficulty and feedback, in a graded manner when forming global self-performance estimates.

In the Discussion, it would be helpful to refer to specific figure panels that correspond to each data-based claim, e.g. line ~377.

Thank you, we have now referred to the relevant figures where appropriate.

Thank you for your review.

REVIEWERS' COMMENTS:

Reviewer #1 (Remarks to the Author):

The authors thoroughly amended their manuscript and addressed all the points I raised.

Reviewer #2 (Remarks to the Author):

I am satisfied by the answers of the authors. They thoroughly answered my questions

Reviewer #3 (Remarks to the Author):

The authors responded adequately to previous comments and my assessment remains positive.